# Insulin/Glucose-Responsive Cells Derived from Induced Pluripotent Stem Cells: Disease Modeling and Treatment of Diabetes

**DOI:** 10.3390/cells9112465

**Published:** 2020-11-12

**Authors:** Sevda Gheibi, Tania Singh, Joao Paulo M. C. M. da Cunha, Malin Fex, Hindrik Mulder

**Affiliations:** Unit of Molecular Metabolism, Lund University Diabetes Centre, Jan Waldenströms gata 35, Box 50332, SE-202 13 Malmö, Sweden; tania.singh@med.lu.se (T.S.); joao_paulo.da_cunha@med.lu.se (J.P.M.C.M.d.C.); malin.fex@med.lu.se (M.F.)

**Keywords:** adipocyte, β-cells, diabetes, hepatocyte, insulin resistance, iPSC, myotube

## Abstract

Type 2 diabetes, characterized by dysfunction of pancreatic β-cells and insulin resistance in peripheral organs, accounts for more than 90% of all diabetes. Despite current developments of new drugs and strategies to prevent/treat diabetes, there is no ideal therapy targeting all aspects of the disease. Restoration, however, of insulin-producing β-cells, as well as insulin-responsive cells, would be a logical strategy for the treatment of diabetes. In recent years, generation of transplantable cells derived from stem cells in vitro has emerged as an important research area. Pluripotent stem cells, either embryonic or induced, are alternative and feasible sources of insulin-secreting and glucose-responsive cells. This notwithstanding, consistent generation of robust glucose/insulin-responsive cells remains challenging. In this review, we describe basic concepts of the generation of induced pluripotent stem cells and subsequent differentiation of these into pancreatic β-like cells, myotubes, as well as adipocyte- and hepatocyte-like cells. Use of these for modeling of human disease is now feasible, while development of replacement therapies requires continued efforts.

## 1. Introduction

The prevalence of diabetes, a life-threatening disease, is increasing worldwide; 463 million people have diabetes and 374 million exhibit impaired glucose tolerance. It has been estimated that 578 million people by the year 2030, and 700 million by the year 2045, will suffer from diabetes [1]. Type 2 diabetes (T2D), which accounts for 90–95% of all diabetes, is essentially characterized by pancreatic β-cell dysfunction and insulin resistance [2]. Therefore, restoration of insulin-producing β-cells, as well as insulin-responsive cells, is a logical therapeutic strategy not only for type 1 diabetes (T1D) but also for T2D.

In recent years, generation of transplantable glucose-responsive, insulin-secreting β-like cells, and insulin-responsive cells in vitro is not only theoretically attractive but is also becoming feasible. Pluripotent stem cells (PSC), such as embryonic stem cells (ESC) and induced PSCs (iPSC), are potential alternative sources of insulin- and glucose-responsive cells owing to their ability to differentiate into all major somatic cell lineages and their unlimited renewal capacity (reviewed in [3,4,5,6]). To this end, PSCs exposed to various growth factors and signaling molecules at specific doses and in a particular sequence, typically mimicking embryonic development, results in successful differentiation into glucose- or insulin-responsive cells [7,8,9]. In addition to replacement therapeutic strategies, and presently more feasible, differentiated PSCs may serve as valuable platforms for drug discovery and elucidation of disease mechanisms in diabetes research, both as in vitro and in vivo model systems [10,11].

Although both ESCs and iPSCs are PSCs, for several reasons iPSCs may be considered as the first choice for modeling and treatment of diseases. There are ethical concerns regarding the use of human embryos for research purposes. In addition, genetic manipulation will likely be required to introduce specific mutations when using ESCs as a model of a genetic disorder. In contrast, iPSCs can be derived from individuals carrying a specific mutation or genetic variant that may be of pathogenetic relevance (for more detail see [12]). For therapy, autologous transplantation of patient-derived iPSC obviates need for immunosuppression although some toxicity may be expected [13,14].

For these reasons, this review will focus on the generation of iPSCs from somatic cells of healthy individuals or patients suffering from various disorders related to T2D and subsequent differentiation of such cells into pancreatic β-like cells, as well as myotubes, and adipocyte- and hepatocyte-like cells. The limitations and challenges of successful therapeutic application of iPSC-derived cells in diabetes, such as development of methods to substitute these cells for host cells, standardization of the treatment protocols, and quality control will also be discussed.

## 2. Reprogramming of Somatic Cells into iPSCs

iPSCs are generated through reprogramming of somatic cells into an embryonic-like state. This is achieved by transduction of pluripotency-associated transcription factors. This was first shown by use of four transcription factors, KLF4 (Kruppel-like factor-4), OCT-3/4 (octamer-binding transcription factor-3/4), SOX2 (sex determining region Y-box 2), and c-MYC, which can reprogram mouse fibroblast into PSCs, exhibiting morphological and molecular features resembling those of ESCs [13]. The initial efforts were focused on use of retroviral vectors [13] and constitutive lentiviruses [15], as well as inducible lentiviruses [16], for reprogramming. A retroviral strategy which can infect only dividing cells, is however associated with a risk of tumor formation, mainly due to reactivation of the c-MYC transgene [17,18]. On the other hand, iPSCs generated without c-MYC virus show decreased tumor frequency but also a reduced efficiency of iPSC generation [19]. To overcome this, another member of MYC, L-MYC, has been used, possessing higher efficiency of iPSC generation but lower tumorigenic activity [20]. Lentivirus, unlike retrovirus, can infect both dividing and non-dividing cells. However, there are concerns regarding incorporation of lentiviral vector sequences into the iPSC genome. Although modifications have been made to the viral-based systems, such as Cre-deletable [21] or inducible lentiviruses [16], use of viral vectors during iPSC generation still lacks the safety required for therapeutic applications.

Transposon-based methods have been applied to remove integrated transgenes from iPSCs. A polycistronic plasmid harboring four factors and a piggyBac transposon have been constructed and integrated into the genome in the presence of piggyBac transposase [22,23]. The inserted fragment is removed by re-expressing transposase following reprogramming. This method demonstrates an efficiency equivalent to retroviral transduction, excises integrated sequences without genome alteration, eliminates the need for viral transduction, and thus serves to create therapeutically usable virus-free iPSCs (reviewed in [24]).

To reduce the limitation of integrative methods, novel non-integrative approaches have been developed for reprogramming purposes. These are divided into four main categories, integration-defective viral delivery (e.g., adenovirus and Sendai viral vectors), episomal delivery, RNA delivery, and protein delivery. Using adenovirus was one of the first attempts to generate integration-free iPSCs. This allows for transient expression of exogenous genes without integration into the genome. Viruses are instead lost by dilution via cell division [25]. The efficiency of this method is, however, low. While the Sendai RNA virus is capable of transducing a wide range of cell types [26], ~10 cell-passages are required to eliminate the virus from reprogrammed iPSCs. To overcome this problem, temperature-sensitive Sendai viruses have been developed to remove viruses by culture at 38 °C [27]. As an alternative, reprogramming based on direct delivery of non-replicating (e.g., plasmids and minicircles) or replicating (e.g., oriP/EBNA1) episomal vectors is now available [28]. Serial transfection utilizing one or two plasmids expressing the key reprogramming factors can generate iPSC lines free of plasmid integration [20,29]. However, due to the multiple transfections, it may be difficult to control the dose of plasmid that the cells receive over the reprogramming period. Larger plasmids have lower transfection efficiency, which results in fewer cells receiving the appropriate dose of plasmid. Furthermore, the plasmid is diluted faster in actively proliferating cells, leading to downregulation of reprogramming factors [28]. Minicircle expression vectors, which are supercoiled DNA molecules lacking a bacterial origin of replication, show higher transfection efficiency and longer-term transgene expression compared to plasmids [30].

To circumvent the problem of non-replication episomal vectors, oriP/EBNA1 vectors have been developed. This approach successfully generates iPSC colonies and one-third of them are free of plasmid DNA [31]. The reprogramming efficiency of episomal vectors, however, is low, but the efficiency can be improved by suppressing p53 and substituting non-transforming L-MYC for c-MYC [32].

Utilizing mRNA to express reprogramming factors has high efficiency, which can be even further improved by adding Lin28 to the Yamanaka reprogramming factors [33]. Culturing at 5% O_2_ with addition of valproic acid in the medium also enhances efficiency [34]. Furthermore, reprogramming factors directly administered as proteins is yet another approach to successfully generate iPSCs. Here, proteins are delivered into cells, fused with specific peptides that mediate the transduction [35]. This method is, however, technically difficult. Generation of recombinant proteins requires fusion of carboxy termini of four reprogramming factors (e.g., HIV transactivator of transcription and poly-arginine domains) [28,36]. The recombinant transcription factors penetrate and cross the plasma membrane of somatic cells within ~six hours and iPSCs can be obtained after four repeated protein transductions [35].

Numerous small molecules and soluble factors can increase the reprogramming efficiency through several mechanisms, such as inhibition of transforming growth factor-β (TGF-β) [37] or Rho-associated, coiled-coil containing protein kinase (ROCK) [38] signaling pathways, inhibition of histone deacetylation [39], induction of glycolysis [40], and increasing functionality of epigenetic modifiers [41]. As chromatin remodeling is a rate-limiting step in the reprogramming of somatic cells, small molecules that alter chromatin modifications (e.g., hydroxamic acid, trichostatin A, and valproic acid) [42] or DNA methylation (e.g., 5′-azacytidine) [42] increase reprogramming efficiency.

Depending on the starting cell types and species, the kinetics of reprogramming are different. This can be due in part to the epigenetic memory in iPSCs; during the reprogramming process, the genes responsible for cell specificity may remain under-reprogrammed [43]. In addition, a variety of genetic and epigenetic aberrations can occur in iPSCs lines. These aberrations along with retained epigenetic markers of somatic cells cause differences in the epigenomes and transcriptomes of iPSCs [44]. The residual signature of epigenomes in iPSCs, known as epigenetic memory, is likely to influence properties of iPSCs.

In summary, several methods have been developed to generate iPSCs. Some hurdles still remain, such as undesirable tumor formation and variable efficiency of reprogramming of different somatic cells (keratinocytes, fibroblasts, hepatocytes, etc.) [17,45]. Therefore, the method of obtaining iPSCs needs to be considered carefully to avoid these issues.

## 3. iPSC-Derived Pancreatic β-Like Cells

### 3.1. Development of the Pancreas

Development of the pancreas is, in general, very similar in different vertebrate species. Since ethical considerations limit studies and use of human embryos, knowledge about pancreas development has largely been derived from the study of mouse pancreatic embryogenesis. During gastrulation, the primitive gut tube (foregut, midgut, and hindgut) is formed from definitive endoderm. The foregut is then split to the anterior and posterior foregut [46]. Transient contact of the notochord with the pre-pancreatic endoderm inhibits expression of the critical transcription factor sonic hedgehog (SHH), resulting in evagination of the posterior foregut and formation of the dorsal and ventral pancreatic buds [47]. In the pancreatic buds, the transcription factors pancreatic and duodenal homeobox 1 (PDX1), SRY (sex-determining region Y)-box 9 (SOX9), and GATA-binding protein 4 (GATA4), are necessary for pancreas growth [48]. Due to the gut rotation, the pancreatic buds then fuse to form the definitive pancreas. There are two developmental phases: the primary transition is characterized by extensive proliferation of pancreatic progenitor cells and formation of stratified epithelium [46,49], and also endocrine cells, which are mainly polyhormonal, expressing both glucagon and insulin, but also some monohormonal cells [50]. In the secondary transition, proliferation of endocrine progenitors and generation of all islet endocrine cells occurs [46].

Several transcription factors and signaling molecules initiate and regulate pancreatic development (Appendix A) by the activation or inhibition of basic signaling pathways such as Wnt (wingless-related integration site)/β-catenin, SHH, and Notch. Co-expression of key transcription factors, such as PDX1, pancreas transcription factor (PTF1A), FOXA2, SOX9, NKX6.1, and carboxypeptidase 1A, regulates differentiation of the multipotent progenitor cells into exocrine, endocrine, and ductal lineages of the pancreas [51,52]. Downregulation of *Ptf1a* and continued expression of *Nkx6.1* lead to the development of ductal/endocrine lineages, while the exocrine lineage depends on the maintenance of *Ptf1a* and loss of *Nkx6.1* [52]. PDX1 in concert with other transcription factors, such as neurogenin 3 (NGN3), NKX6.1, and MAFA, promotes specification and maturation of multipotent progenitor cells into pancreatic β-cells [7,53].

### 3.2. Differentiation of Pancreatic β-Like Cells from iPSC

Protocols to generate glucose-responsive pancreatic β-cells from iPSCs largely follow strategies established for ESCs (Table 1). They are designed to mimic pancreatic organogenesis by sequential treatment of iPSCs with specified growth and differentiation factors in a chemically defined medium. Most protocols are multi-stage including: (a) induction of definitive endoderm, (b) formation of primitive tube, (c) development of posterior foregut, (d) development of progenitor cells, (e) production of immature pancreatic β-cells, and (f) mature β-like cells [7,9,11,14,54] (Figure 1). Numerous small and large molecules have been used to promote β-cell differentiation from iPSCs (Table 2). Transgenic expression of pancreas-specific transcription factors such as FOXA2, PTF1A, PDX1, hepatocyte nuclear factor (HNF) 4A, HNF6, NGN3, PAX4, NEUROD1, NKX6.1, and MAFA is used to evaluate the differentiation efficiency [7,9,11,14,54].

#### 3.2.1. Induction of Definitive Endoderm

Formation of definitive endoderm is the rate-determining step in pancreatic differentiation [55]. It is based on mimicking the actions of TGF-β, Wnt, and nodal signaling by inhibiting the phosphoinositide 3-kinase (PI3K) and glycogen synthase kinase-3β (GSK-3β) signaling pathways. Use of activin A, a member of the TGF-β superfamily, and also Wnt, for induction of undifferentiated iPSCs into definitive endoderm, are important elements that are common to most methods [14,54]. Activin A, mimicking nodal action, promotes iPSCs differentiation into SOX17^+^ definitive endoderm. For a response to activin/Nodal, PI3K signaling must be suppressed. For this purpose, compounds such as wortmannin and LY2994002, which inhibit PI3K signaling, have been found to promote definitive endoderm formation [56]. Another signaling molecule, which appears to modify the activity of activin A during the definitive endoderm induction step, is the TGF-β superfamily molecule, bone morphogenetic protein 4 (BMP4) [48]. Inclusion of low concentrations of BMP4 along with activin A in the first day of differentiation has been shown to improve the efficiency of definitive endoderm formation [14]. Indeed, a combination of activin A, BMP4, and fibroblast growth factor 2 (FGF2) induces PSC-differentiated endoderm more efficiently than activin A alone [57,58]; it can be even more efficient in the presence of a ROCK inhibitor under serum-free conditions [59].

Inhibition of GSK-3β (using CHIR99021) for induction of definitive endoderm and spheroid formation at the final stage are important for the generation of functional iPSC-derived β-like cells [60]. CHIR99021, in the presence of BMP4, promotes endodermal cell viability and increases the SOX17^+^ cells rate. Transplantation of these cells into streptozotocin (STZ)-induced diabetic mice reduces blood glucose levels over the next 4 weeks [60]. Expression of pluripotency markers (NANOG, POU5F1, and SOX2) drops during the first days of iPSCs differentiation, while expression of the mesoendodermal stage-specific marker and then the definitive endoderm-specific markers SOX17 and C-X-C chemokine receptor 4 (CXRC4) is increased [11].

#### 3.2.2. Formation of Primitive Gut Tube and Development of Posterior Foregut

SHH inhibitors and FGF10 prime definitive endoderm to form cells characteristic of the primitive gut tube. At low concentrations, FGF promotes posterior foregut fate and expression of *PDX1*, while at higher concentrations it promotes more posterior endoderm fates [61,62]. Members of the FGF family, especially FGF7/keratinocyte growth factor (KGF) and FGF10, are commonly used before PDX1 induction to promote formation of the primitive gut tube which is capable of generating the pancreatic epithelium [9,53,57]. Addition of FGF inhibitors, such as SU5402, LY294002, and U1026, blocks the formation of PDX1^+^ progenitors [63]; therefore PDX1^+^/NKX6.1^+^ progenitors cannot be developed in the absence of FGF7 [64]. Furthermore, inclusion of FGF4 to the differentiation culture during the retinoic acid-mediated PDX1 induction step enhances *PDX1* expression, most likely through increases in progenitor cell survival [65]. SHH acts as an anti-pancreatic factor, as forced expression of SHH inhibits development of the pancreas [66]. Thus, its inhibition in the region of the primitive gut tube gives rise to the pancreas and this is essential for pancreatic specification. The SHH inhibitor, cyclopamine is routinely used during the retinoic acid induction step [11,67]. At this stage, definitive endoderm cell markers are downregulated, while expression of *HNF1B* and *HNF4A* is increased [11]. There are other pathways which may play a regulatory role at this step as inclusion of indolactam V, a strong activator of protein kinase C (PKC), raises *PDX1*, *NGN3*, *NKX2.2* and *NKX6.1* expression following retinoic acid treatment [14,67,68].

#### 3.2.3. Development of Progenitor Cells

Pancreatic progenitor cells express a group of transcription factors, of which PDX1 and NKX6.1 are critical markers for β-cell maturation and functionality (for more detail see [69,70]). PDX1^+^/NKX6.1^+^ progenitors differentiate into monohormonal β-cells, while PDX1^+^/NKX6.1^−^ progenitors differentiate into polyhormonal cells [71,72]. The differentiation efficiency of iPSCs to PDX1^+^/NKX6.1^+^ progenitors is high under optimized conditions [70,71,73]; the PDX1^+^/NKX6.1^−^ population is further increased when duration of the posterior foregut stage is prolonged [71]. Although the differentiation efficiency of PDX1^+^/NKX6.1^+^ progenitors is reasonably stable, using the same protocol on different iPSC lines leads to a variable NKX6.1 induction, ranging from 37% to 84% [74]. This indicates that the differentiation of pancreatic progenitors/β-cells also depends on inherent differences across cell lines. Recently, PDX1^−^/NKX6.1^+^ progenitor cells have been found during differentiation of iPSCs to β-like cells [75]; these progenitor cells have similarities to a subset of the pancreatic mesenchymal stem cells (MSC) that can give rise to INS^+^ cells. PDX1^−^/NKX6.1^+^ progenitors demonstrate downregulation of pancreatic epithelial genes and upregulation of neuronal development genes, indicating that they represent a unique source for generating INS^+^ cells of a non-epithelial origin [75].

Expression of NKX6.1 is promoted by use of nicotinamide and EGF, which increase generation of pancreatic progenitors [74]. Additionally, YAP, a member of the Hippo signaling pathway, is involved in progenitor specification and differentiation into functional pancreatic endocrine cells [76]. The Hippo pathway integrates tissue architecture by balancing between progenitor self-renewal and differentiation [77]. YAP expression is downregulated late in NKX6.1^+^ progenitors and persists upon completion of the differentiation, as ~95% of endocrine progenitors and insulin-expressing β-cells do not express YAP. Inhibition of YAP during the specification of early PDX1^+^ to late NKX6.1^+^ progenitors decreases the number of NKX6.1^+^ progenitors, while its inhibition during endocrine specification, leads to differentiation of pancreatic progenitors into NGN3^+^ endocrine progenitors and then into NKX6.1^+^/C-peptide^+^ β-cells more efficiently than control cells [76]. Transplantation of iPSC-derived pancreatic PDX1^+^/NKX6.1^+^ progenitor cells into diabetic mice reverses hyperglycemia [78].

Forskolin (an adenylate cyclase activator), activin receptor-like kinase (ALK5) inhibitors, ROCK inhibitors, γ-secretase inhibitor XXI, nicotinamide, triiodothyronine (T3), exendin-4, heparin, and dexamethasone also enhance the generation of insulin-expressing cells from PDX1^+^ progenitors [14,54,79]. FGF10 acts upstream of notch signaling in pancreatic progenitor cell proliferation; exogenous FGF10 enhances proliferation of pancreatic progenitor cells and promotes the expression of *HES1*, a downstream target of notch signaling [80]. *Fgf10*-deficient mice exhibit reduced proliferation of pancreatic epithelial progenitor cells and display pancreatic hypoplasia [81]. In contrast, overexpression of *Fgf10* leads to pancreatic hyperplasia, decreased differentiation of endocrine progenitor cells, and increased progenitor cell numbers [82].

Inhibition of Notch signaling following induction of the PDX1^+^/NKX6.1^+^ progenitors leads to the generation of NGN3^+^ endocrine progenitors. The transcription factor NGN3 plays an important role at this stage as impaired Notch receptor activation or signaling upregulates *NGN3* expression, resulting in premature endocrine cell differentiation. NGN3 is expressed early in the specification of the pancreatic endocrine lineage [83]. Null mutations in *NGN3* are associated with neonatal diabetes and blocking of β-cell differentiation [83], as well as complete loss of all types of pancreatic endocrine cells in *Ngn3*-deficient mice [84]. Inhibition of BMP and TGF-β/Activin A/nodal is required for efficient endocrine development [57]; blocking these pathways increases insulin expression and total number of cells. Abrogation of BMP signaling seems to be responsible for increased insulin expression, while TGF-β/Activin A/nodal inhibition enhances cell numbers [57]. High concentrations of Noggin (to inhibit BMP4) are essential for inducing differentiation, first into PDX1^+^ progenitors and then into NGN3-expressing pancreatic endocrine progenitors [67]. Differentiation may progress to the stage of insulin expression even without inhibition of BMP [79]. This can be due in part to the low level of endogenous BMP in iPSC lines used for pancreatic differentiation. It generates insulin-positive cells without requiring inhibition of this pathway. At this stage, expression *of NKX6.1* is increased, demonstrating the specification of endocrine pancreas. A significant portion of endodermal chromogranin A^−^/PDX1^+^ cells also expresses *NKX6.1* [11].

**Figure 1 cells-09-02465-f001:**
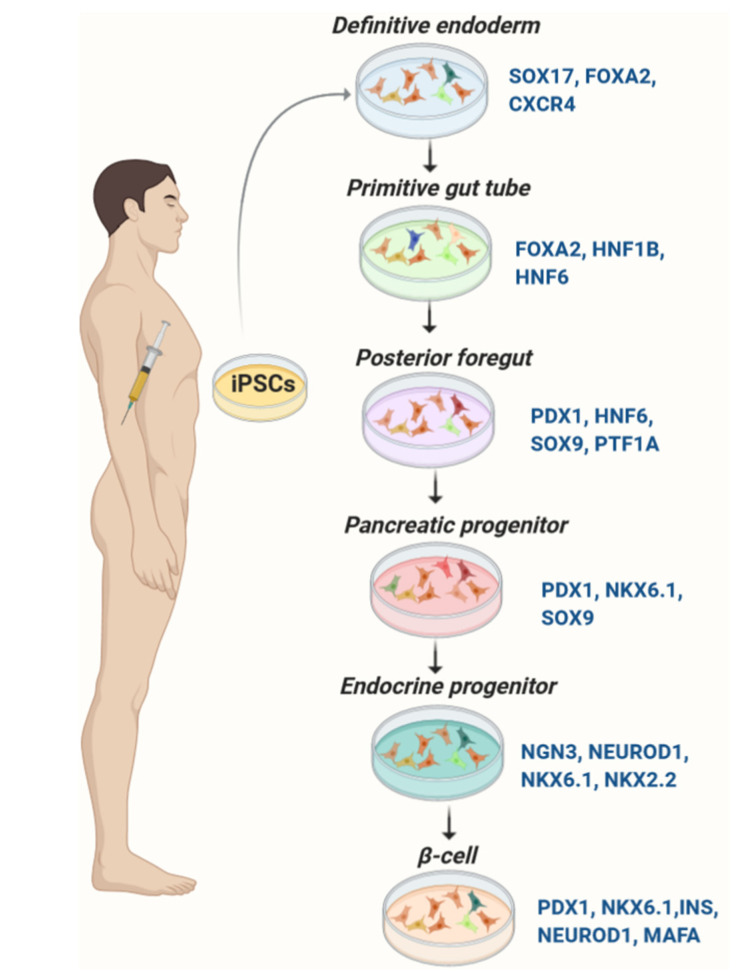
Differentiation of pancreatic β-cells from iPSCs. Expression of the key transcription factors is monitored for evaluation of the consecutive stages of differentiation.

#### 3.2.4. Production of Pancreatic β-Like Cells from iPSCs

Immature fetal β-cells require several weeks to gain responsiveness to glucose [85]; this inability may be due to the lower expression levels of the key β-cell transcription factors controlling the insulin secretory machinery. For example, expression of transcription factors MAFA, NEUROD1, NKX6.1, and PDX1 in the islets of neonatal rats is lower prior to that in islets that have acquired glucose-stimulated insulin secretion (GSIS). MAFA and insulin gene expression reach adult levels by three months [86], concomitant with the ability of β-cells to sense glucose [85]. MAFA binds to the insulin MAF-responsive element and activates insulin gene expression in response to glucose [87]. Additionally, overexpression of *MAFA* increases expression of glucokinase, glucose transporter *GLUT2*, *PDX1*, *NKX6.1*, glucagon-like peptide-1 receptor (*GLP1R*), and pyruvate carboxylase [88], as well as enhancing functional maturation of immature β-like cells derived from PSCs (for more details refer to [89]).

The differentiation of the endocrine progenitors into β-like cells is promoted by vitamin E, T3, betacellulin, heparin, ALK5 inhibitors, AXL receptor tyrosine kinase inhibitors, exendin-4, hepatocyte growth factor (HGF), insulin-like growth factor 1 (IGF-1), and N-acetyl cysteine, which induce the expression of *MAFA* [9,14,53,54,67,90,91]. Inclusion of T3 during the last stages of differentiation increases the proportion of cells co-expressing *NKX6.1, PDX1,* insulin, and *NEUROD1*, likely through increasing *MAFA* and *NGN3* expression. 

Using step-wise differentiation protocols, several groups have successfully generated glucose-responsive iPSC-derived β-like cells. Glucose sensing and the amount of secreted insulin are, however, not equivalent to that observed in cadaveric human islets in vitro. The insulin secretion capacity of iPSC-derived β-cells is qualitatively similar to that of cadaveric islets in terms of the ability to regulate de- and re-polarization of the plasma membrane in response to KCl or altering ATP-sensitive potassium channel (K_ATP_ channel) activity [90,92,93]. iPSC-derived pancreatic endodermal cells also exhibit a Ca^2+^ response and increased insulin secretion following glucose stimulation [11]. Compared to adult human primary β-cells, this glucose-stimulated Ca^2+^ increase is heterogeneous, demonstrating both oscillatory and biphasic kinetics [11]. Plasma membrane depolarization by KCl causes an immediate and transient Ca^2+^ increase in iPSC-derived β-cells [11]. These observations indicate that K_ATP_ channels and voltage-gated Ca^2+^ channels in these β-like cells are functional.

Low levels of secretion of insulin in response to glucose may be due to constraints of metabolism. The capacity of iPSC-derived β-cells to transport glucose across the plasma membrane is similar to that in cadaveric islets [92]. iPSC-derived β-cells also demonstrate an increase in oxygen consumption rate (OCR) after a glucose challenge but they are not able to maintain fully uncoupled OCR, and rapidly lose electron transport chain activity [92]. Differentiated β-like cells contain equal or even higher amounts of mitochondrial mass per cell compared to cadaveric islets, suggesting that β-like cells may have metabolically dysfunctional mitochondria or that there is an insufficient metabolic substrate supply to sustain the electron transport chain during glucose challenge. Accumulation of the early glycolysis-derived metabolites dihydroxyacetone phosphate, glycerol-3-phosphate, and phosphoenolpyruvate is similar in cadaveric islets and iPSC-derived β-cells [92]; the most striking differences between β-like cells and cadaveric islets appear in mitochondrial metabolite pools owing to anaplerotic generation of oxaloacetate. The total phosphoenolpyruvate metabolite pool derived from glycolysis and oxaloacetate is much smaller in β-like cells than in cadaveric islets [92]. GSIS in iPSC-derived β-like cells is limited by the enzymatic conversion of glyceraldehyde-3-phosphate to 3-phosphoglycerate, which is catalyzed by glyceraldehyde 3-phosphate dehydrogenase (GAPDH) and phosphoglycerate kinase (PGK1). Although there is no difference in protein levels of GAPDH and PGK1, the activities of both enzymes are reduced in β-like cells [92].

Estrogen-related receptor γ (ERRγ) has been suggested as a master regulator of β-cell maturation [94]. Postnatal induction of ERRγ expression promotes mitochondrial oxidative phosphorylation, electron transport chain activity, and ATP production, all of which are essential for GSIS [94]. Overexpression of ERRγ in iPSC-derived β-like cells triggers a metabolic transformation that facilities GSIS in vitro. These differentiated cells reduce blood glucose levels when transplanted into diabetic mice [94]. Additionally, testosterone has been shown to promote the differentiation efficiency of iPSC-derived β-like cells by increasing expression of pancreatic β-cell progenitor master genes [95].

Pretreatment of mesenchymal stromal cells with epigenetic modifiers has been shown to increase expression of insulin, GLUT2, glucokinase, as well as the transcription factors PDX1, NKX6.1, and MAFA [96]. The differentiated β-like cells are glucose-responsive and show a significant decrease in global DNA methylation level [96]. In addition, ROCKII inhibition at the pancreatic progenitor stage promotes the maturation of iPSC-derived β-cells by increasing the percentage of INS^+^ cells at the end of differentiation. Inhibition of ROCKII enhances NKX6.1, INS, UCN3, MAFA and G6PC2 expression levels, improves glucose sensitivity, as well as reduces expression of cell cycle and focal adhesion genes, conferring robustness to functionally mature iPSC-derived β-cells [97].

Using the differentiation protocol described by Ameri et al. [98], our group has successfully generated β-like cells from iPSC lines derived from patients with T2D. Using a six-stage protocol, the iPSCs are driven from a single layer cell in the undifferentiated state through all developmental stages, finally forming clusters of β-like cells at day 35 (Figure 2A–G). To confirm β-cell differentiation, insulin gene expression was determined at different time points of the differentiation, showing a marked increase from day 29 and onward (Figure 2H). Furthermore, the clusters exhibited strong immunostaining for insulin and PDX1, but much less so for glucagon (Figure 2I–L).

Although in vitro iPSC differentiation protocols do not generate pure populations of β-like cells, some cell surface markers can be used to isolate pure and functional iPSC-derived β-cells and re-aggregate them before encapsulation for transplantation. Cell surface markers SUSD2, CD200, and CD318 identify endocrine progenitors (NGN3^+^ cells) [64]. Glycoprotein 2 (GP2), CD142, and CD24, have been shown to identify β-cell precursors [99,100,101]. Cell surface marker CD49a identifies β-like cells [102]. Currently, the final step of β-cell maturation is accomplished in vivo after transplantation into mammalian hosts [54]. The molecular mechanisms of in vivo terminal differentiation are not fully understood. Different potential scenarios, like the involvement of neuronal signals [103], circulating factors [104,105], and the impact of three dimensional (3D) niches [106] have, however, been implicated.

**Figure 2 cells-09-02465-f002:**
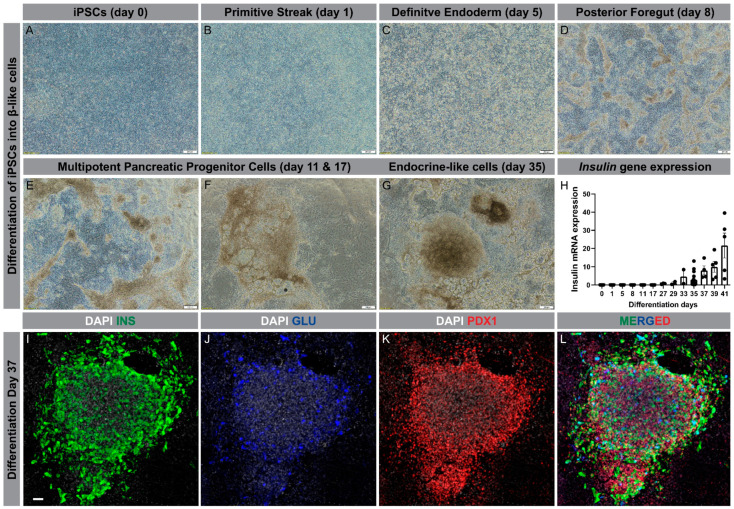
Differentiation of human iPSCs into β-like cells. (**A**–**G**): Progression through different developmental stages during differentiation of iPSCs into β-like cells (scale bar is 200 µm); (**H**): Insulin gene expression through the course of differentiation, ranging from day 0 to day 41; (**I**–**L**): Condensed cluster of differentiated cells at day 37 displaying protein expression of insulin (**green**), glucagon (**blue**), and PDX1 (**red**); nucleus (**grey**) (Scale bar is 50 µm).

#### 3.2.5. Impact of Cellular Microenvironment in Differentiation of iPSC-Derived β-Cells

Pancreatic islet architecture with intra-islet cell-cell communication between β-cells and other endocrine cell types are essential for β-cell functionality and glucose responsiveness. During islet development, for proper β-cell lineage specification, differentiation and maturation, intact cell-cell (via intercellular adherents junctions: cadherins, connexins, and nectins) and cell-matrix (via mechanical cues through extracellular matrix (ECM) stiffness, topography, fluid shear forces, and geometry) interactions are crucial for coordinating and supporting cells [107,108,109,110,111,112]. The pancreas is considered a soft organ based on its ECM composition. This favors a low range of physiological stiffness for β-cell development in vivo, which is contrary to the standard 2D monolayer differentiation culture systems, possessing a much higher stiffness than the native islet niches [111]. Observations from culture of human islets reveal that soft substrates maintain islet cells coalescence and organization, while stronger extracellular-cell interactions in stiffer culture conditions cause cells to scatter and lose their function [113].

Success in producing and maintaining functional β-cells in vitro greatly depends on the ability to recapitulate the native islet 3D microenvironment, with its spatial, chemical and mechanical attributes. The 3D microenvironment of the islet niche is composed of multiple layers of interconnected cells, nerve fibers, blood vessels, and ECM molecules (such as fibronectin, vitronectin, laminins, and collagens). All together, they serve to generate synchronized insulin secretion [112,114,115,116,117,118,119]. Vascular endothelial cells release soluble factors that play a crucial role for endocrine cell development and proper glucose-responsive insulin release from β-cells [120]. Furthermore, by providing a basal membrane, vascular endothelial cells also are responsible for cell–cell interactions between α- and β-cells [120].

Alterations of the ECM are sensed by β-cell mechanosensors (such as primary cilia, ion channels, integrins, glycocalyx, and cadherins), which transduce signaling cascades intracellularly. Thus, cytoskeletal dynamics and formation of integrin adhesion complexes are brought into motion [107,110]. These signals of mechanotransduction modify cellular programs by altering the nuclear architecture, gene transcription via mechanosensitive transcription factors (YAP/TAZ, TCF/LEF, etc.), mitochondrial dynamics, and the RhoA/ROCK pathway [107,121]. Aggregated (confined) cell clusters exposed to laminin are more robust in producing pancreatic progenitors expressing PDX1 and NKX6.1, as well as NGN3^+^ endocrine progenitors, rather than spread out loose cells, reflecting a more ductal phenotype [122]. This phenomenon is accompanied by the appearance of reduced formation of stress fibers and mediated by actin polymerization and YAP-Notch mechanosignaling [122].

To provide 3D cell-cell interactions, suspension/aggregate/spinner flask culture systems have been developed to induce formation of islet-like organoids, which further promote formation of glucose-responsive β-cells from stem cells [78,102,123,124]. By use of such a system, monohormonal and glucose-responsive β-like cells were produced that show key features of bona fide β-cells, like high mRNA levels of the transcription factor MAFA and β-cell ultrastructure [90]. β-cells generated in 3D culture system and primary adult islets show similar responses to multiple sequential high glucose challenges and depolarization with KCl. Notably, some iPSC-derived β-like cells exhibit insulin content and secretion at levels similar to that of primary islets; both the entire population, as well as individual iPSC-derived β-like cells, show Ca^2+^ fluxes similar to those in primary human islet cells [90].

Although 3D culture systems are considered superior to 2D systems, recently, differential actin cytoskeleton states (polymerized vs. depolymerized), cell adhesion intensity to the surface of culture plates, and substrate stiffness have been shown to control the timing of *NEUROG3*, and *NKX6.1* expression, hereby inducing pancreatic endocrine lineage [125]. A 2D planar differentiation protocol was established by including a chemical compound (latrunculin A), which specifically alters cytoskeletal dynamics [125]. This enhances the differentiation efficiency and functionality of ESCs, as well as iPSC-derived β-like cells. This demonstrates that use of 3D culturing systems, in order to produce glucose-responsive iPSC-derived β-like cells, can be substituted by manipulating cell-biomaterial interactions with an efficient 2D planar protocol, hereby opening new opportunities to improve differentiation outcomes [125].

#### 3.2.6. iPSC-Derived β-Like Cells: Treatment of Diabetes and Disease-in-a-Dish Models

Generation of β-cells from PSCs holds great promise in the treatment of diabetes. In addition, diabetes patient-derived iPSCs could serve as experimental models to investigate disease mechanisms. There are more than 400 genetic signals associated with T2D and fasting glucose [126]. Many of these “diabetes risk loci” may perturb β-cell number, and deregulate proinsulin conversion and insulin secretion (for details refer to [127]). iPSCs from patients with either T1D [128] or T2D [129] have been established. We also have generated several iPSC lines from patients carrying T2D risk alleles and successfully differentiated β-like cells from these iPSC lines. For therapeutic purposes, generating β-cells from a patient’s own iPSCs would potentially also limit immune reaction and hence the need for immune suppressants (at least in T2D). Such drugs have various side effects, which may interfere with insulin action and sensitivity, for instance, steroids. However, immunogenicity of iPSCs characterized by T-cell infiltration and massive necrosis after transplantation in syngeneic recipients has been reported [130]; this can be due in part to aberrant antigens resulting from long-term maintenance or immunogenic alteration due to rapid in vitro differentiation. Although many differentiation protocols have been established on feeder-free culture systems, some protocols still use numerous animal-derived products that may exert unknown effects on cells, increase immune reactions and risks of graft rejection (reviewed in [131]).

Encapsulation of iPSC-derived β-cells could solve issues with immunity: a flat-sheet macroencapsulation device, to graft the iPSC-derived pancreatic endoderm cells under the skin of immune-compromised mice, increases circulating C-peptide levels in response to glucose administration [11]; both fasting and glucose-stimulated C-peptide levels continue to rise in response to glucose after 18 weeks post-implantation, indicating that the maturation of iPSC-derived β-cells is furthered over time in vivo [11]. Typical grafts contain multiple cell clusters individually expressing insulin, glucagon, somatostatin, and ghrelin in a heterogeneous cellular architecture, reminiscent of either fetal or more or less immature pancreatic islets [11]. Encapsulation of pancreatic endocrine progenitors enhances the rate of hormone-positive and insulin-expressing cells co-expressing key β-cell markers and therefore promotes the differentiation outcome. Encapsulation at the first differentiation stages improves early differentiation signals, whereas encapsulation at a later differentiation stage improves expression of hormones and factors involved in hormone synthesis and secretion [9]. Additionally, transplantation of iPSC-derived pancreatic endodermal cells in STZ-induced diabetic mice increases glucose clearance for several weeks after treatment, while removal of the implants renders animals strongly hyperglycemic [11]. Transplantation of iPSC-induced β-like cells under the kidney capsule of immunodeficient diabetic mice normalizes blood glucose levels after two weeks [90]. In addition, analysis of C-peptide and glucagon staining reveals that these β-cells remain monohormonal after transplantation [90].

**Table 1 cells-09-02465-t001:** Overview of protocols used for differentiation of pancreatic β-like cells from human iPSCs.

iPSC Source	Protocol	In Vivo/Vitro Efficacy	Stage/Day	Ref.
Human iPSCs	Days 1–3: GDF8 and GSK3β inh. Days 4–5: FGF7 and VitC. Days 6–7: FGF7, VitC, 1 µM RA, SANT, TPB, LDN. Days 8–10: FGF7, VitC, 100 nM RA, SANT, TPB, LDN. Days 11–13: 50 nM RA, T3, SANT, ALK5 inh, LDN. Days 14–21/29: T3, ALK5 inh, LDN, γ-secretase inh. Days 21/29–28/36: T3, ALK5 inh, AXL inh, N-Acetylcysteine	↑ GSIS and plasma human c-peptide levels after transplantation in mice	7/28–36	[78]
Human iPSCs	Days 1–2: Activin A, CHIR99021. Day 3: No feed. Day 4: KGF. Day5: No feed. Day 6: KGF. Days 7–8: LDN, KGF, SANT, Y-27632, RA, PdBU. Days 9–14: KGF, SANT, Y-27632, RA, Activin A. Days 15–22: Betacellulin, RA, T3, ALK5 inh, SANT, Heparin, γ-secretase inh. Days 22–29/36: T3, ALK5 inh	↑ GSIS in vivo and in vitro	6/29–36	[132]
Human iPSCs	Days 1–2: Activin A, CHIR99021. Days 2–4: Activin A. Days 4–7: KGF. Days 7–9: KGF, SANT1, RA, LDN (only Day 7), PdBU. Days 9–14: KGF, SANT, RA. Days 14–18: RA, SANT1, ITS-X, VitC, Heparin, T3, ALK5 inh, Betacellulin, γ-secretase inh. Days 18–21: RA, ITS-X, VitC, Heparin, T3, ALK5 inh, Betacellulin, γ-secretase inh. Days 21–35: ALK5 inh, T3	↑ GSIS in vitro	6/35	[92]
Human iPSCs	Days 1–2: Activin A, Wnt3a, VitC. Days 3–4: Activin A. Days 5–12: dorsomorphin, RA, SB431542 (TGFβ inh). Days 13–23: Forskolin dex, ALK5 inh, Nicotinamide, T3. Days 24–27/29: overexpression of ERRγ	↑ KCl stimulated insulin secretion without ERRγ overexpression. ↔ GSIS without ERRγ overexpression. ↑ KCl and glucose stimulated insulin secretion after ERRγ overexpression	3/23	[94]
Human iPSCs	Days 1–3: Activin A, CHIR99021. Days 4–7: KGF. Days 8–9: KGF, SANT1, RA, LDN, PdBU. Days 10–15: KGF, SANT, RA. Days 16–23: RA, SANT1, T3, ALK5 inh, Betacellulin, γ-secretase inh. Days 18–21: RA, ITS-X, VitC, Heparin, T3, ALK5 inh, Betacellulin, γ-secretase inh. Days 22–60: ALK5 inh, T3	↑ GSIS in vitro	6/22–60	[102]
Human iPSCs	Days 1–2: Activin A, CHIR99021. Days 2–3: Activin A. Days 4–5: VitC, KGF. Days 6–10: Insulin, ITS-X, KGF, SANT, RA, LDN, TPB	NR	4/10	[7]
Human iPSCs	Days 1–3: CHIR99021, GDF8, Activin A, Wnt3A. Days 4–5: FGF7, VitC. Days 6–10: FGF7, VitC, RA, TPB, LDN, Noggin, SANT. Days 11–13: RA, LDN, SANT, ALK5 inh, T3. Days 14–25: LDN, ALK5 inh, T3, γ-secretase inh, Heparin. Days 30–45: ALK5 inh, T3, Cyclopamine, AXL inh.	NR	7/45	[9]
Human iPSCs	Day 0: Wnt3A, Activin A, Y-27632, ITS-X. Day 1: Activin A, Y-27632, ITS-X. Day 2: ITS-X, KGF, ALK4 inh, Y-27632. Day 3: ITS-X, KGF, ALK4 inh. Day 4: KGF. Days 5–7: TTNBP, Noggin, Cyclopamine, Heregulin. Days 8–12: Noggin, Heregulin, EGF, KGF, Y-27632	↑ Fasting and glucose-stimulated human c-peptide levels in vivo	4/12	[11]
Human iPSCs	Days 1–3: Activin A, CHIR99021. Days 4–5: FGF7, VitC. Days 6–7: FGF7, VitC, RA, TPB, LDN, SANT. Days 8–10: FGF7, VitC, RA, LDN, SANT, EGE, Nicotinamide. Days 11–14: LDN, ALK5 inh, Betacellulin, Heparin, RA, ITS-X, GC1, ZnSO4. Days 14–28: LDN, ALK5 inh, Betacellulin, Heparin, γ-secretase inh, ITS-X, GC1, ZnSO4. Days 29–35: ALK5 inh, Heparin, ITS-X, GC1, ZnSO4, Trolox, JNK inh, Resveratrol, N-Acetylcysteine, AXL inh	↑ GSIS and plasma human c-peptide levels after transplantation in mice	7/35	[133]
Human iPSCs	Days 1–3: Activin A, CHIR99021, Y-27632, DMSO. Days 3–7: KGF. Days 7–10: KGF, Noggin, VitC, TTNPB, Cyclopamine. Days 10–14: KGF, EGF, Nicotinamide, Y-27632, VitC. Days 14–17: SANT-1, RA, ALK5 inh, LDN, T3, γ-secretase inh, bFGF, XAV939 (Wnt inh), Y-27632	↓ Fasting blood glucose and ↑ plasma human c-peptide after transplantation into diabetic mice	5/17	[54]
Human iPSCs	Days 1–3: CHIR99021, GDF8. Days 4–5: FGF7, VitC. Days 6–7: FGF7, VitC, RA, TPB, LDN, Noggin, SANT, ITS-X. Days 8–10: ITS-X, Heparin, RA, LDN, SANT, ALK5 inh, T3. Days 11–13: ITS-X, SANT, LDN, ALK5 inh, T3, RA, Heparin. Days 14–20: ALK5 inh, T3, LDN, γ-secretase inh. Days 20–27: ALK5 inh, T3, N-Acetylcysteine, AXL inh. Day 28: Wnt 3/4/5 or Wnt inhibitor (G007-LK) for 4h. Days 28–30/37: ALK5 inh, T3, N-Acetylcysteine, AXL inh	↔ GSIS ↔ KCl stimulated insulin secretion	7/37	[53]
Human iPSCs	Days 0–1: Activin A, B27, CHIR99021. Days 1–3: B27, Activin A. Days 3–10: B27, RA, Dorsomorphin, SB431542. Days 10–20: B27, Forskolin, Repsox, Nicotinamide, Dex, Testosterone	↔ GSIS ↑KCl stimulated insulin secretion	3/20	[95]
Human iPSCs	Days 1–2: Activin A, 2-ME, CHIR99021. Days 3–5: Activin A, 2-ME. Days 6–7: 2-ME, Cyclopamine, FGF10. Days 8–13: Noggin, RA, Cyclopamine, ALK5 inh, 2-ME. Days 14–15: 2-ME, Noggin, AlLK5 inh, Indolactam V. Days 16–23: 2-ME, Exendin4, Nicotinamide, IBMX, Forskolin	↑ c-peptide secretion in the presence of KCl, K_ATP_ channel blocker, LVDCC and muscarinic agonists	5/23	[67]
Human iPSCs	Days 1–2: Activin A, CHIR99021, FGF2, BMP4, 2-ME. Days 3–4: KSR, Activin A, 2-ME. Days 5–7: FGF7, ITS-X. Days 8–11: FGF7, ITS-X, SANT-1, LDN, EC23, Indolactam V. Days 12–14: FGF10, ITS-X, SANT-1, LDN, EC23, Indolactam V. Days 15–21: EGF, ITS-X, SANT-1, LDN, EC23, ZnSO4, Indolactam V, RepSox, Heparin, Nicotinamide, Exendin4, Y-27632, γ-secretase inh. Days 22–31: BMP4, HGF, IGF, ITS-X, ZnSO4, Indolactam V, RepSox, Heparin, Nicotinamide, Exendin4, Forskolin	↓ Nonfasting blood glucose, improved glucose tolerance, ↑ plasma human c-peptide after transplantation into diabetic mice	6/31	[14]
Human iPSCs	Days 1–2: Activin A, CHIR99021. Days 2–4: Activin A. Days 4–7: KGF. Days 7–9: KGF, SANT1, RA, LDN (only Day 7), PdBU. Days 9–14: KGF, SANT, RA. Days 14–18: RA, SANT1, ITS-X, VitC, Heparin, T3, ALK5 inh, Betacellulin, γ-secretase inh. Days 18–21: RA, ITS-X, VitC, Heparin, T3, ALK5 inh, Betacellulin, γ-secretase inh. Days 21–35: ALK5 inh, T3	↑ GSIS both in vitro and in vivo	6/35	[90]
Human iPSCs	Days 1–5: Activin A, Wnt3a. Days 5–7: KGF, VitC, Y27632. Days 7–8: KGF, VitC. Days 8–12: SANT-1, RA, Noggin, TPB, VitC, KGF. Days 12–16: ALK5 inh, Noggin, GLP-1, SANT-1, RA, γ-secretase inh, Heparin, T3. Days 16–17: ALK5 inh, Noggin, GLP-1, γ-secretase inh, Heparin, T3. Days 17–27: Nicotinamide, IGF-1, GLP-1, ALK5 inh, T3, Heparin	↑ GSIS and KCl stimulated insulin secretion both in vivo and in vitro	5/27	[91]

PdbU, phorbol 12,13-dibutyrate; TTNPB, tetrahydro tetramethyl naphthalenyl propenyl benzoic acid; KSR, knockout serum replacement; KGF, keratinocyte growth factor; ITS-X, insulin-transferrin-selenium-ethanolamine; LVDCC, L-type voltage-dependent Ca^2+^ channel; RA, retinoic acid; Dex, dexamethasone; HGF, hepatocyte growth factor; GLP-1, glucagon-like peptide-1; IGF-1, insulin-like growth factor 1; ALK inh, activin receptor-like kinase inhibitor; TPB, PKC activator; T3, triiodothyronine; GC1, thyroid hormone receptor-β agonist; AXL inh, receptor tyrosine kinase inhibitor; 2-ME, 2-Mercaptoethanol; IBMX, isobutylmethylxanthin; BMP4, bone morphogenetic protein 4; GSIS, glucose stimulated insulin secretion; NR, Not reported; ↑ increased, ↓ decreased, ↔ no effect.

## 4. iPSC-Derived Insulin-Responsive Cells and Insulin Resistance

### 4.1. Insulin Resistance

Insulin resistance, a key component of T2D pathophysiology is defined as a state with decreased metabolic actions of insulin in target tissues, namely liver, skeletal muscle, and adipose tissue [2]. Rare defects in the insulin receptor reduce insulin sensitivity, while the more frequent post-receptor perturbations reduce its effects [2]. Environmental factors associated with physical inactivity and obesity account for insulin resistance in the great majority of individuals.

This notwithstanding, longitudinal studies have shown that insulin resistance is also a heritable trait and develops in people at risk for T2D many years before glucose intolerance [134,135]. As with potential β-cell dysfunction, a significant proportion of the genetic variants associated with T2D are likely to be implicated in target tissue dysfunction and hence insulin resistance [126]. For this reason, and others, iPSCs derived from insulin-resistant patients are a unique tool to identify molecular mechanisms of this metabolic dysfunction. It also allows testing of drugs and development of cell-based therapies. Indeed, all insulin target cells, such as skeletal myotubes, adipocytes, and hepatocytes can be generated from patient-specific iPSCs. By carrying the same genetic signature as T2D patients, they may help to identify the genetic factors involved in the disease progression.

### 4.2. iPSC-Derived Hepatocytes

#### 4.2.1. Development of the Liver

Hepatocytes, the principal liver cell type, are derived from embryonic endoderm [136]. Liver-specific transcriptional factors and external signaling events originating from the cardiac mesoderm result in differentiation of the endoderm to hepatoblasts, then to the liver bud, and eventually a fully formed liver. *FOXA* proteins regulate almost all liver-specific genes [137]. Cardiogenic mesoderm, by expressing different FGF proteins, provides critical extracellular signals to liver progenitor cells [136,137]. The septum transversum mesenchyme also expresses BMP2 and BMP4 and plays a role before and during the induction of hepatoblasts within the endoderm. Hematopoietically expressed and Prospero-related homeobox factors, HEX and PROX1, through encoding transcriptional regulatory proteins, as well as ECM proteins that interact with β1 integrin receptors, play an important role in liver bud development [136].

#### 4.2.2. Differentiation of iPSCs into Hepatocytes

A combination of growth factors and small molecules is used for differentiating hepatic developmental stages from definitive endoderm to immature hepatocytes (hepatic progenitors) and finally mature hepatocytes (Table 3) [138,139].

##### Differentiation of Definitive Endoderm and Immature Hepatocytes

For differentiation of definitive endoderm, various signaling pathways controlling the early cell fate decisions of pluripotent stem cells, such as those involving Wnt, Activin, BMP, and FGF are activated. Following generation of endodermal cells from iPSCs, hepatoblasts are formed. To this end, HGF or KGF is added. At this stage, differentiated hepatoblasts display expression of FOXA1, FOXA2, CCAAT enhancer-binding protein alpha (CEBPα), peroxisome proliferator-activated receptor alpha (PPARα), alpha-fetoprotein (AFP), and cytokeratin (KRT)-18 and -19 [72,140,141,142,143]. After treatment with HGF, cells begin to express the adult isoforms of HNF1α and HNF4α, liver X receptor (LXR), ATP-binding cassette transporter A1 (ABCA1), alpha-1-antitrypsin (A1AT), albumin, as well as liver-specific microRNAs miR122, miR148a, and miR194 [72]. In addition, expression of retinoid X receptor (RXR) and vascular endothelial growth factor receptor (VEGFR) is increased and then declines progressively towards the final stages of differentiation [72].

**Table 2 cells-09-02465-t002:** Small molecules used for differentiation of insulin/glucose-responsive cells from iPSCs.

Molecule	Function	Ref.
Activin A	Member of the TGF-β family; induces DE lineage from stem cells	[144]
ALK5iII/RepSox	TGFβR-1/ALK5 inhibitor, upregulates expression of UCN3, MAFA, NKX6.1, and PDX1; induces hepatic and myocyte differentiation	[8,79]
B27	Supports cell growth, viability and induction of endoderm lineage. Promotes β-cell differentiation, maturation, and increases the number of insulin^+^ cells. Also induces hepatocyte differentiation	[98,145,146]
Betacellulin	EGF receptor ligand. Maintains expression of NKX6.1 and PDX1 in endocrine progenitors. Aids in inducing MAFA expression in β-like cells	[147]
bFGF/FGF2	Suppresses SHH signaling and initiates pancreatic differentiation by inducing PDX1 expression. Also induces hepatoblasts and myogenic differentiation by deriving mesoderm lineage	[63,72,148]
CHIR99021	Inhibits GSK3α/β and promotes Wnt signaling for efficient induction of DE lineage	[79,149]
Cyclopamine	Blocks SHH signaling. Induces PE lineage and promotes PDX1 expression	[150,151]
Db-cAMP	Nerve growth factor. Induces expression of MAFA and insulin	[152]
Dexamethasone	Enhances β-like cell differentiation and proliferation. Increases the number of insulin^+^ cells at the end of differentiation. Aids maturation of hepatocytes and adipocytes	[72,79,153]
DMSO	Can be used in combination with activin A to stimulate the process of DE induction	[154]
Dorsomorphin	BMP inhibitor, enhances PDX1^+^ expression in the PE stage. Aids in maturing β-like cells	[79,152]
EC23	Synthetic retinoic acid receptor agonist, used for the formation of PE	[14]
EGF	Stimulates cell growth, differentiation and maturation of several cell types including hepatocytes, myocytes and β-cells. Expands the PDX1^+^ PPs and promotes endocrine cell fate	[155]
Exendin-4	Analog of GLP1. Promotes β-like cell maturation by enhancing the expression of GCK, GLUT2, and NEUROD1	[156]
Fasudil	RhoA/Rho kinase (Rock) inhibitor; promotes DE	[157]
FGF10	Aids in the induction of DE and enhances the characteristic markers of PE	[158]
FGF4	At high concentration promotes endodermal cell fate and expansion	[65]
FGF7	Induces expression of PDX1, PTF1A, and HLXB9. Aids in producing 3D cellular clusters	[159]
Forskolin	Increases the levels of cAMP. Derives differentiation and maturation of hepatocytes, myocytes and β-cells. Is required for priming β-cell differentiation and insulin expression	[8,79,160]
GDF8	Belongs to the TGFβ family and induces DE	[78]
Glutamine	Induces myocytes characteristics while differentiating. Induces hepatic specification from DE lineage	[140,161]
Heparin	Co-factor for FGF2. Enhances generation of endocrine cells and mature β-like cells from PDX1^+^ PPs	[133]
Heregulin	Member of the EGF family used in deriving PE cells	[11]
HGF	Matures β-like cells, hepatocytes, and myocytes	[151,162]
Hydrocortisone	Matures hepatocytes while differentiating from DE lineage	[163]
IBMX	Phosphodiesterene inhibitor and an adenosine receptor antagonist. Induces adipocyte differentiation and maturation. Enhances insulin expression and proportion of differentiating β-like cells	[164]
IDE1/2	Activator of the SMAD2/3 pathway and induces DE lineage	[165]
IGF-1	Induces myogenic as well as β-like cell differentiation and maturation	[166]
Indolactam V	Activator of the PKC pathway; induces PDX^+^ PPs	[167]
ITS-X	Supports differentiation and maturation of hepatocytes and adipocytes. Also aids in formation of PPs to insulin-producing β-like cells	[72,168,169]
KGF/FGF7	Generates PDX1^+^ PPs and PDX1^+^/NKX6.1^+^ endocrine progenitor cells. Drives hepatoblasts from foregut endoderm cells	[8,170]
LDN	BMP type 1 receptor inhibitor. Promotes PDX1^+^ PPs and maturation of β-like cells	[171]
LY294002	Inhibits GSK3-β and PI3K activity for efficient induction of DE lineage	[144]
Lysophosphatidic acid	Acts through G protein-coupled receptors. Induces hepatoblast differentiation and expansion	[8]
N-acetyl cysteine	Functions as an antioxidant. Enhances expression of MAFA	[145,172]
NECA	Activates adenosine signaling and promotes β-like cell proliferation	[164]
Nicotinamide	A poly (ADP-ribose) synthetase inhibitor; promotes expression of PDX1 up to the later stages in β cell differentiation process. Crucial for hepatocyte differentiation, proliferation and maturation	[147,173]
Noggin	BMP inhibitor, induces PDX1^+^ PPs and NGN3^+^ endocrine progenitors by suppressing hepatic lineage differentiation	[67]
Oncostatin M	Member of IL-6 cytokine family and is crucial for liver development in the final stage of hepatocyte differentiation	[174]
PdBU	A phorbol ester, acts as an activator of PKC and is used in promoting pancreatic differentiation	[90]
Pioglitazone	An antidiabetic drug, induces lipid-accumulating adipocyte differentiation	[175]
Plasmanate	A plasma protein fraction used for inducing adipocyte differentiation	[176]
Resveratrol	A stilbenoid polyphenol, enhances the expression of key β-cell maturation genes	[177]
Retinoic acid	Crucial for generating NGN3^+^ endocrine progenitors and for the β-cell specification. Differentiates hepatoblasts into cholangiocyte progenitors. Depending on its concentration and stage administration, it can have a variable but crucial effect on adipocyte differentiation	[72,178,179,180]
RG108	Inhibits DNA methyltransferase, Stimulate reprogramming from somatic cells to iPSCs	[39]
RKI-1447	Rho-kinase inhibitor, induces DE lineage and aids differentiation into PDX1^+^ PPs	[157]
Rosiglitazone	An antidiabetic drug, derives adipogenesis by enhancing the expression of PPARγ and C/EBP-α as well as activation of MAPK and PI3K pathways	[181]
SANT-1	SHH signaling inhibitor, enhances formation of PE and PDX1^+^ NKX6.1^+^ PPs	[74]
SB431542	TGF-β receptor inhibitor, enhances number of NKX6.1^+^ NGN3^+^ endocrine progenitors	[145]
Sodium Butyrate	Inhibits histone deacetylation and aids in DE lineage induction	[182]
Sodium cromoglicate	Enhances NGN3^+^ endocrine precursors and insulin^+^ cells	[183]
Sphingosine-1-phosphate	A signaling sphingolipid metabolite, aids hepatoblast expansion during differentiation	[8]
Stauprimide	Belongs to the family of indolocarbazoles, derives DE lineage by downregulating c-Myc expression	[184]
Triiodothyronine (T3)	Induces MAFA expression and generates mono-hormonal insulin^+^ cells. Induces and maintains brown/beige adipogenesis	[104,145,164]
Taurine	Induces PE lineage, promotes insulin expression in β-like cells	[185]
Thiazovivin	Rho-kinase inhibitor, induces DE lineage	[157]
TPB	A PKC activator, enhances generation of NKX6.1^+^ PPs and endocrine progenitors	[78]
TTNPB	Analog of retinoic acid, aids in pancreas specification	[54]
Vitamin C	Induces PDX1^+^ NKX6.1^+^ PPs and prevents the formation of polyhormonal cells during β-cells differentiation. Also induces expression of hepatocyte-specific genes and aid in its maturation process. Induces mesoderm lineage in adipocyte differentiation	[78,186,187]
Wortmannin	Inhibits GSK3-β and PI3K activity and induces DE lineage	[149]
XAV939	Tankyrase inhibitor which targets Wnt/β signaling and promotes β-like cell maturation	[53]
Y27632	Inhibitor of ROCK, enhances PPs and supports cluster formation	[124]

ALK5Iii, Activin receptor-like kinase 5 inhibitor II; bFGF, Basic fibroblast growth factor; Db-cAMP, Ascorbic acid Dibutyryl-cyclic AMP; DMSO, Dimethyl sulfoxide; EGF, Epidermal growth factor; GDF8, Growth Differentiation Factor 8; HGF, Hepatocyte growth factor; IBMX, Isobutyl methylxanthine; IDE1/2, Definitive Endoderm 1/2 inducer; IGF-1, Insulin-like growth factor 1; ITS-X, Insulin-Transferrin-Selenium-Ethanolamine; KGF, Keratinocyte growth factor, NECA, N-Ethylcarboxamidoadenosine; PdBU, Phorbol dibutyrate; SANT-1, Sonic hedgehog agonist-1, TPB, Trifluoromethyl phenyl pentadienoylamino benzolactam; TTNPB, Tetrahydro tetramethyl naphthalenyl propenyl benzoic acid; DE, Definitive Endoderm; PPs, Pancreatic Progenitors; PE, Pancreatic Endoderm.

##### Mature Hepatocyte Differentiation

Oncostatin M (OSM) is important for hepatic maturation [174]. Oncostatin M which belongs to interleukin (IL)-6 group of cytokines is produced by hematopoietic cells. In addition, various chemical compounds and small molecules such as dimethylsulfoxide (DMSO), dexamethasone, hydrocortisone-21- hemisuccinate and Ile-(6) aminohexanoic amide (dihexa) are used for iPSC differentiation from definitive endoderm to mature hepatocytes [72,140,141,143]. These molecules regulate specific target(s) in signaling and epigenetic mechanisms, as well as manipulate cell fate without genetic alterations. Expression of HNF1α and HNF4α, LXR, A1AT, ABCA1, CEBPα, albumin, liver-specific microRNAs miR122, miR148a, and miR194 peaks in mature hepatocytes (Figure 3) [72]. Some markers such as tryptophan-2,3-dioxygenase (TDO) [141], tyrosine amino-transferase (TAT), CEBP-β, specific cytochrome P450 superfamily, liver-specific arginase-1 (involved in the production of urea), and asialoglycoprotein receptor 1, as well as liver-specific enzymes, such as uridine diphosphate glucuronosyl transferase 1 A1 (UGTA1) and fumarylacetoacetate hydrolase (FAH), are assessed to identify mature hepatocytes [72,140,141,143].

**Figure 3 cells-09-02465-f003:**
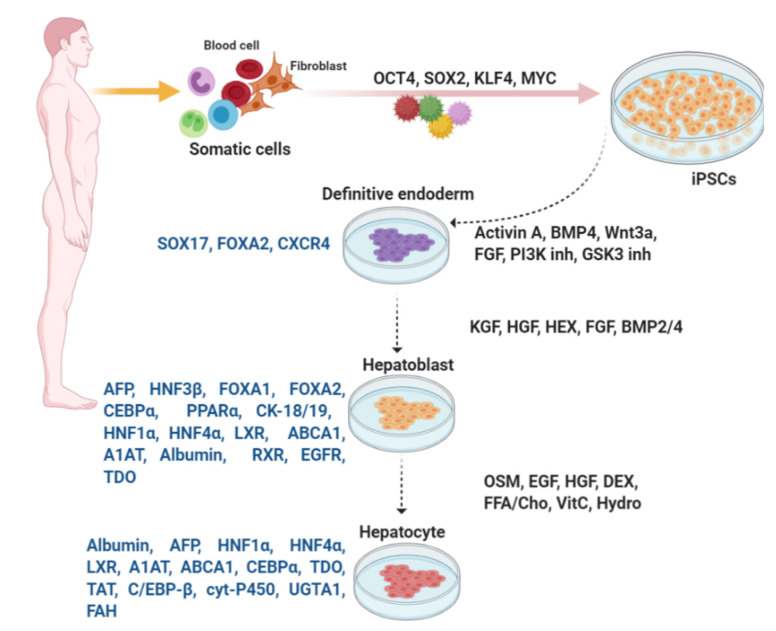
In vitro differentiation of hepatocytes from iPSCs. Similar to differentiation of β-like cells, embryonic development is mimicked by addition of numerous large or small molecules to induce each stage of differentiation. Expression of the key transcription factors is monitored for evaluation of the consecutive stages of differentiation.

#### 4.2.3. Functional Evaluation of iPSC-Derived Hepatocytes

iPSC-derived hepatocytes secrete albumin [8,10,72], store lipids [10], accumulate glycogen [8,10], produce urea [8,10], and eliminate ammonia [8]. They also express liver transcription factors and asialoglycoprotein receptor (specific for differentiated hepatocytes), absorb low-density lipoprotein (LDL) [188], express cytochrome p450s (3A4 and 7A1) [8,10], as well as E-Cadherin [10]. As with β-like cells, most iPSC-derived hepatocytes resemble more fetal or neonatal cells [189], as they have lower expression of enzymes involved in metabolic functions and drug metabolism [190]. The cytochrome P450s of the liver play critical roles in the maintenance of lipid homeostasis and detoxification of endogenous compounds and drugs. Recently, use of ECM components, such as collagen, laminin, and fibronectin, may mimic the liver microenvironment and allow cells to arrange themselves in a more physiologically relevant manner [191,192]. A 3D culture system can also be used to increase the efficiency of differentiation.

Moreover, genes involved in cell-cycle and cell-proliferation pathways have significantly higher expression in fetal than adult liver. In contrast, genes highly expressed in adult liver are metabolism-related genes, including three major functional categories: metabolism of fatty acids, xenobiotics, and glucose [193]. To mimic these metabolic changes, low glucose concentration media during the last four days of differentiation with added free fatty acids, bile acids, cholesterol, corticoids, epidermal growth factor (EGF), and rifampicin, promote further hepatocyte maturation of human iPSCs [72]. Maturation in the organ-like microenvironment is associated with increased expression of genes encoding insulin signaling/lipogenesis and mature, clinically relevant enzymes [72]; there is no quantitative difference in intracellular triglyceride content, mitochondrial number, and expressed levels of mitochondrial DNA [72]. However, cytochrome P450 activity (CYP3A4) in iPSC-derived hepatocytes after exposure to rifampicin is at the level of human fetal hepatocytes but inferior to that in adult human hepatocytes [72]. 

iPSC-derived hepatocytes have been used to resolve inherited metabolic disorders of the liver, such as A1AT deficiency, familial hypercholesterolemia, and glycogen storage disease type 1a [194]. Gene editing of an α1-antitrypsin-deficient iPSC line restores the structure and function of α1-antitrypsin in the derived hepatocytes both in vitro and in vivo [195].

**Table 3 cells-09-02465-t003:** Overview of protocols used for differentiation of hepatocytes from various sources of iPSCs.

iPSC Source	Protocol	In Vivo/Vitro Efficacy	Stage/Day	Ref.
Human iPSCs	Days 1–3: Wnt3A, Activin A. Days 4–5: Activin A. Days 5–8: KSR, DMSO. Days 9–14: HGF, OSM, Hyd	Glycoproteins, glycogen, and lipid production. secretion of albumin, urea, AFP, and A1AT; cytochrome P450 1A2 and 3A4 activities	3/14	[10]
Human iPSCs	Day 1: Wnt3A, Activin A, BMP4, bFGF. Days 2–3: Activin A, BMP4, bFGF. Days 4–5; KGF, ALK inh. Days 6–9; KGF, BMP2, BMP4, bFGF. Days 10–16/18: Forskolin, ALK inh, EGF, LPA, Dex, S1P, GSK3β inh. Days 17/19–36/38: ALK inh, Forskolin	Glycogen production; secretion of albumin, urea, AFP; cytochrome P450 3A4 activities; ammonia elimination	6/36–38	[8]
Human iPSCs	Days 1–2: BMP4, Activin A, FGF2. Days 3–4: Activin A. Days 5–9: BMP4, FGF2. Days 10–13: Activin A, FGF10, RA. Days 14–18: EGF, Dex, FFA, Hyd, Nicotinamide, IL-6, TGFβ1, VitC, ITS-X, sDLL-1	Glycogen, albumin, urea, and A1AT production; cytochrome P450 3A4 activities, intracellular triglyceride content	4/18	[72]
Human iPSCs	Days 0–3: Activin A. Days 4–7: BMP2, FGF4. Days 8–13: HGF, KGF. Days 14–18: OSM, Dex. Days 19–21: OSM, Dex, N2B27	Glycogen, albumin, and urea, production; cytochrome P450 activity	5/21	[196]
Human iPSCs	Days 1–3: B27^−^, Sodium butyrate, Wnt3a, Activin. Days 4–5/6: B27, Wnt3a, Activin. Days 6/7–9/10: KSR, DMSO, Glutamine, 2ME. Days 9/10–11/12: iPSCs were mixed with MSCs and HUVECs and cultured within OSM, Transferrin, Hyd, VitC, Insulin, GA-1000, Bovine brain extract, hEGF, FBS, HGF, Dex	Cellular polarity and bile acid transport; urea production and glycogen accumulation	3/12	[140]
Mouse iPSCs	Days 1–5: Activin A, Wnt3a. Days 6–10: BMP4, FGF-2. Days 11–15: HGF. Days 16–20: HGF, OSM, Dex, ITS-X	Urea and albumin production	4/20	[141]
Human iPSCs	Days 1–3: B27, Activin A, Wnt3a. Days 4–10: DMSO, KSR. Days 11–20: HGF, OSM, Hyd	Urea and albumin production	3/20	[143]

KSR, knockout serum replacement; Hyd, Hydrocortisone; LPA, lysophosphatidic acid; S1P, sphingosine-1-phosphate; ITS-X, Insulin-transferrin-selenium-ethanolamine; OSM, Oncostatin M; HGF, Hepatocyte growth factor; FGF, Fibroblast growth factor; BMP4, Bone morphogenetic protein 4; Dex, dexamethasone; KGF, keratinocyte growth factor; ALK inh, activin receptor-like kinase inhibitor; RA, retinoic acid; 2-ME, 2-Mercaptoethanol; EGF, Epidermal growth factor; IL-6, Interlukin-6; AFP, Alpha-fetoprotein; A1AT, Alpha-1-antitrypsin.

### 4.3. iPSC-Derived Skeletal Muscle Cells

#### 4.3.1. Development of Skeletal Muscle

Nearly all skeletal muscle cells are derived from paraxial mesoderm, forming somites, then dermamyotome and finally the myotome [197]. For myotome development, muscle progenitor cells delaminate from the four edges of the dermomyotome; these progenitor cells also migrate into the limb buds. Paired box protein3 (PAX3) as well as cMET, a tyrosine kinase receptor that binds HGF, are important for this delamination and migration [198]. *Pax3* is then downregulated and the delaminating progenitor cells differentiate into myoblasts [198]. The expression of myogenic factor 5 (*MyF5*), myogenic regulatory factor (*MrF4*), and myogenic differentiation (*MyoD*) is increased in the myoblasts, which differentiate into myocytes through the action of myogenin (*MyoG*), *MrF4*, and *MyoD* [197]. The myocytes fuse and mature into multinucleated muscle fibers forming a continuous muscle layer, the myotome. Signaling molecules such as SHH, Wnt proteins, and BMPs are involved in the developmental processes [5,197]. 

During the early embryonic/primary phase, primary myofibers are generated, which derive from PAX3^+^ or PAX3^+^/PAX7^+^ dermomyotomal progenitors [5]. These myofibers, which form the early myotomes and limb muscles, express a specific set of proteins, such as myosin light chain 1 (MYL1) and slow myosin heavy chain (MYH). In the fetal/secondary phase, expression of *Pax7* and downregulation of *Pax3* are initiated in a subset of PAX3^+^ myogenic progenitors. These PAX7^+^ myogenic precursors form primary fibers and then generate the secondary or fetal fibers expressing specific markers, such as β-enolase or MYL3 [5,199]. At this time, the fibers express fast MYH isoforms and muscle growth is continued by cell fusion and the inclusion of myonuclei from proliferating PAX7^+^ progenitors. A portion of the PAX7^+^ progenitors will also generate satellite cells, the pool of adult muscle stem cells [199].

#### 4.3.2. Differentiation of iPSCs into Myocytes

There are two protocols for derivation of skeletal muscle cells from iPSCs: i) forced overexpression of muscle-specific transcription factors, such as PAX3, PAX7, and MYOD, and ii) a step-wise induction of skeletal muscle by small molecules to inhibit or activate relevant signaling pathways in myogenesis (Table 4). 

##### Overexpression of Muscle-Specific Transcription Factors

Activation of *MYOD* in a variety of differentiated cell lines is sufficient to activate a downstream program for terminal muscle differentiation [200]. Different systems of gene expression, such as lentiviral and piggyBac-based approaches, are used to express *PAX7* and *MYOD1* in iPSCs [161,201]. *MYOD1* overexpression, particularly in undifferentiated cells, drives them along the myogenic lineage with 70–90% efficiency, and myocytes reach maturity within two weeks of differentiation [161]. Overexpression of *PAX7* in two well-characterized human iPSC lines, generated from normal donor’s fibroblasts, converts these cells into myotubes [201]. Overexpressing muscle-specific transcription factors secures high efficiency of progenitor preparation and yields progenitors more rapidly [161,201]. Progenitors can also be sufficiently enriched by fluorescence-activated cell sorting (FACS) [201,202]. However, a high level or sustained expression of *MyoD* induces cell cycle arrest [203]. As these methods use exogenous genes, the resulting cells may not fully reflect the normal processes of progenitor proliferation/differentiation/maturation. Furthermore, genetic modification is another concern when progenitors are destined for cell-based therapy in patients. Myogenic progenitors exposed to growth factors and/or signaling molecules may be more suitable for transplantation in patients.

##### Step-Wise Induction of Skeletal Muscle Cells by Small Molecules

Some small molecules secreted as paracrine factors play important roles in muscle development, controlling proliferation, migration, and differentiation from mesodermal cells into somites and dermomyotome [204]. iPSCs are treated with a combination of GSK-3β inhibitor (6-bromoindirubin-3′-oxime (6-BIO) or CHIR99021), adenylyl cyclase activator, and bFGF, which promote myogenic differentiation [148,160]. Administration of CHIR99021 is a critical step, as lower concentrations fail to result in myogenic progenitors and high concentrations or a longer exposure is toxic [169,202,205]. By contrast, 6-BIO demonstrates the lowest toxicity among other GSK-3β inhibitors [206]. Inhibition of GSK-3β promotes Wnt signaling, which in turn promotes cellular differentiation to a mesoderm or endoderm fate [202,205], resulting in up to 90% of myogenic cells, evidenced by exclusive PAX7^+^/MYOG^+^ populations in vitro [148].

FGF2, EGF, IGF-1, HGF, and platelet-derived growth factor (PDGF) have also been used to enhance myogenic differentiation [169,202,205,206,207,208]. DMSO and PI3K inhibitors (LY294002) are also used to increase differentiation efficiency toward the mesodermal lineage in the presence of BMP4 [148]. DMSO enhances the differentiation efficiency of PSCs into terminal cell lineages through activation of the retinoblastoma protein [209]. Likewise, PI3K inhibition increases the differentiation of stem cells into mesodermal lineages through the promotion of activin A and nodal signaling [144]. During embryonic development, morphogen gradients of BMPs, Wnt, FGF, and retinoic acid control somitogenesis and eventual separation of paraxial, intermediate, and lateral plate mesoderm [204]. BMP4, in tandem with FGF and Wnt signaling, potentiates development of the paraxial mesoderm along the primitive streak into the dermomyotome, which forms the skeletal musculature of the body [204,210]. 

After 7, 12, and 36 days, cells are differentiated into embryoid bodies (EBs), myoblasts, and myotubes, respectively [160]. From day 17 onwards, cells proliferate rapidly, cultures reaching complete confluency after 24 days; multinucleated myotube-like cells are observed between 30 and 40 days [169]. At day 36, up to 64% of nuclei express the late-stage skeletal muscle transcription factor MYOG, indicating highly efficient differentiation toward a skeletal muscle fate [148]. Differentiated iPSCs also have higher expression of *MYOD1*, as well as of *MYH* and the cholinergic receptor *CHRNA1* [148]. Under proliferating conditions, expanded myogenic progenitors (about 25 days) of iPSCs express PAX7 in a subset of cells. Myogenic progenitor cultures contain stable PAX7^+^ cells during the main part of the expansion period [169] (Figure 4). 

For inducing spontaneous twitching in cell culture, serum-free medium is used in terminal differentiation, resulting in robust spontaneous myotube contractions [148,205]. Supplementation of 0.5–2% FBS increases the overall survival of the culture but also the proliferation rate of mononucleated cells, resulting in overgrowth of the cell culture [169], while supplementation with serum free-medium results in fibers with fast MYH, titin, and α-actinin, as well as striation patterns contracting spontaneously. 

**Figure 4 cells-09-02465-f004:**
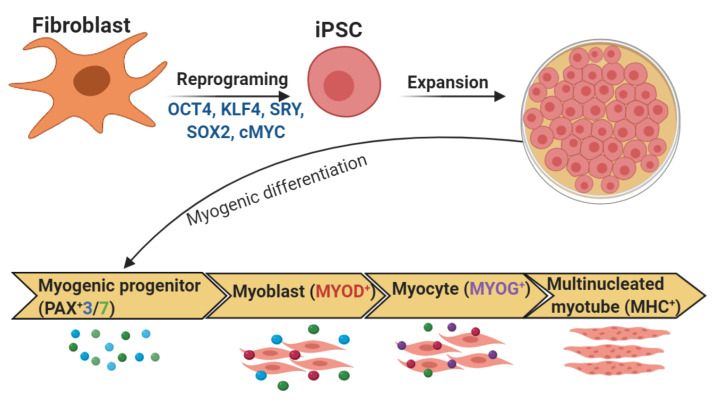
In vitro differentiation of myotubes from iPSCs. Addition of a number of large or small molecules induces each stage of differentiation, which is monitored by expression of key transcription factors during the consecutive stages of differentiation.

#### 4.3.3. iPSC-Derived Skeletal Muscle Cells and Insulin Resistance

Molecular defects contributing to metabolic dysregulation have been identified in iPSCs-generated from patients with Donohue syndrome [160,211]. Morphologically, these iPSCs are differentiated normally, but both mRNA and protein expression of the insulin receptor are reduced in iPSCs generated from these patients [160,211]. However, the results of insulin-stimulated phosphorylation of downstream signaling are not consistent. Both reduced [160] and no changes [211] in insulin-stimulated phosphorylation of insulin receptor, insulin receptor substrate 1 (IRS1), and Akt have been reported. Moreover, these studies have demonstrated both elevated [211] and reduced [160] insulin-stimulated phosphorylation of IGF1 receptor; extracellular signal-regulated kinase (ERK) phosphorylation tends to be reduced [211], while there is no difference in its protein level [160]. Basal expression of metabolic and growth-regulatory genes, including ras-related associated with diabetes (RAD1), hexokinase 2, and GLUT4 is reduced in iPSCs generated from patients with Donohue syndrome [160].

As insulin is involved in regulating the unique properties of self-renewal and pluripotency, insulin-resistant iPSCs are frequently defective in their ability of self-renewal [212]. Indeed, this circumstance is a major bottleneck in the use of reprogrammed somatic cells for subsequent differentiation to mature cells, regardless of whether they are intended for research or therapy. Indeed, the pathogenetic processes to be examined is also a key factor determining stem cell differentiation. Expression of the early growth response genes is increased in iPSC-derived healthy myotubes in response to insulin, while insulin responsiveness is decreased in insulin-resistant iPSCs [160]. These iPSCs also show higher OCR, higher extracellular acidification, increases in oxidative phosphorylation protein complexes III and V, as well as increased lactate release [211]. Metabolic effects of insulin are mimicked in healthy iPSC-derived myotubes with increased 2-deoxyglucose uptake, glycogen synthase activity, and glycogen accumulation [160]. This is similar in magnitude to the best insulin-responsive murine skeletal muscle cell models. In contrast, insulin-resistant iPSC-derived myotubes demonstrate an almost complete failure of insulin-stimulated, but not basal, glucose uptake, glycogen synthase activity, and glycogen accumulation [160].

Insulin-resistant iPSCs exhibit mitochondrial dysfunction with an increase in mitochondrial number and decrease in size [213]. This dysfunction is associated with elevated expression of mitochondrial fission factor and inverted formin 2, which both enhance mitochondrial fission, while expression of genes regulating mitochondrial fusion are unchanged [213]. Increased mitochondrial fission has also been reported in pancreatic β-cells and skeletal muscle of T2D patients [214]. Moreover, alterations in mitochondrial DNA could be associated with genetic risk factors for T2D [215]. Insulin-resistant iPSCs exhibit distinct phenotypes with elevated lactate levels and decreased citrate synthase activity [213]. Interestingly, increased plasma lactate levels [216,217] and decreased citrate synthase activity have been reported in skeletal muscle [218] and adipose tissue [219] in insulin resistance and T2D. In the iPSCs, differences in citrate synthase activity are reversed after exogenous oxaloacetate administration [213], indicating that availability of substrate might be limiting for tricarboxylic acid (TCA) cycle metabolism in insulin resistance.

Accumulation of glycogen in the myotubes from Pompe patients is reversed in gene-edited iPSC-derived myotubes; *GAA* mRNA expression as well as its enzymatic activity are raised 5.5-fold and 3-fold over levels in healthy control myotubes [169]. This shows the possibility of combining a myogenic differentiation protocol with gene editing in iPSCs to produce gene-corrected skeletal muscle cells.

**Table 4 cells-09-02465-t004:** Overview of protocols used for differentiation of myotubes from various sources of iPSCs.

iPSCs Source	Protocol	In Vivo/Vitro Efficacy	Stage/Day	Ref.
Human iPSCs	Days 1–7: GSK3 inh, bFGF, Forskolin. Days 8–9: Serum free media. Days 10–36: Horse serum	Insulin-stimulated glucose uptake, glycogen synthase activity, glycogen accumulation	3/36	[160]
Human iPSCs	Days 1–5: GSK3 inh. Days 5–19: bFGF, DMSO. Days 20–35: KSR, ITS-X	Glycogen accumulation	3/35	[169]
Human iPSCs	Days −1–0: DMSO. Days 0–1.5: Insulin, transferrin, FGF2, PI3K inh, BMP4, GSK3 inh. Days 1.5–7: Insulin, Transferrin, FGF2, PI3K inh. Days 8–12: 15% FBS. Days 13–36: horse serum	Spontaneous contractions	3/36	[148]
Human iPSCs	Days 1: MyoD overexpression. Day 3: Adding G418. Day 4: ROCK inh. Day 5: ROCK inh, Dox. Days 6–10: αMEM, KSR, 2-ME. Days 11–13: horse serum, IGF1, 2-ME, Glutamin	Fusion potential both in vitro and in vivo	3/13	[161]
Equine iPSCs	Day 1: mTeSR1 media. Days 2–3: ITS-X, GSK3 inh, ALK inh. Days 4–5: ITS-X, bFGF, GSK3 inh, ALK inh. Days 6–7: IGF1, bFGF, HGF, BMP inh. Days 8–12: IGF1, KSR. Days 12–30: IGF1, KSR, bFGF	Intracellular Ca^2+^ release in response to KCl-induced membrane depolarization	5/30	[220]
Equine iPSCs	Days 1–14: media containing high glucose, 10% FBS. Day 15: MyoD overexpression. Day 16: washing with PBS and adding puromycin for selection of positive transductants. Puromycin-resistant cells were cultured in high glucose media containing 10% FBS then Dox was added and cells were differentiated for 7 days.	Intracellular Ca^2+^ release in response to KCl-induced membrane depolarization	?/?	[220]

KSR, knockout serum replacement; ITS-X, Insulin-transferrin-selenium-ethanolamine; HGF, Hepatocyte growth factor; FGF, Fibroblast growth factor; BMP, Bone morphogenetic protein; ALK inh, activin receptor-like kinase inhibitor; 2-ME, 2-Mercaptoethanol; IGF-1, Insulin-like growth factor 1; Dox, Doxycycline.

### 4.4. The iPSC-Derived Adipocyte

#### 4.4.1. Development of Adipocytes

The overall function of white adipose tissue (WAT) is to store energy, whereas brown adipose tissue (BAT) dissipates energy in a heat-producing process called thermogenesis. Postnatally, a third form of adipose tissue has been identified (‘beige’) but its function and origin is enigmatic; it has been reported that beige adipocytes are generated from unique precursor cells [221], but there is also evidence that they may be generated from WAT by transdifferentiation [222]. Adipogenesis is the differentiation of fibroblast-like MSCs into adipocytes. This is generally divided into generation of adipocyte progenitors (or pre-adipocytes) from multipotent MSCs and a differentiation step. Adipogenesis is controlled by several groups of transcription factors, including the *CEBP* gene family, PPARγ, PPARγ coactivator-1α (PGC-1α), and PRD1-BF-1-RIZ1 homologous domain containing protein-16 (PRDM16) [223]. PPARγ, a master regulator of adipogenesis, is crucial for maturation of adipocytes [224]. In addition, PPARγ is a receptor for insulin-sensitizing drugs [225]. Expression of *Cebpa* is important for terminal differentiation of adipocytes, as its absence leads to insulin resistance; development of BAT is independent of *Cebpa* [226]. The expression of PGC-1α is high in BAT, where it upregulates expression of several genes of the TCA cycle and contributes to fatty acid utilization [227]. PRDM16 regulates the differentiation of brown and beige adipocytes as its overexpression in WAT promotes beige adipocyte development [228]; its deletion causes a profound loss of beige adipocyte function in mice [229].

#### 4.4.2. Differentiation of iPSCs into Adipocytes

Two protocols for differentiation of adipocytes from iPSCs have been reported: (i) forced overexpression of white/brown adipocyte-specific transcription factors, such as PPARγ, PRDM16, and CEBPα, and (ii) step-wise induction of adipocytes by small molecules to inhibit or activate relevant signaling pathways in adipogenesis (Table 5).

##### Formation of Embryoid Bodies

Embryoid bodies are three-dimensional cell aggregates formed in suspension by PSCs, which differentiate into cells of all three germ layers. Formation of EBs is a routine inductive step to generate specific cell lineages from PSCs. For formation of EBs, iPSCs are cultured in a floating condition with medium containing undefined components, such as fetal bovine serum (FBS), knock-out serum replacement (KSR) or albumin products for 2–7 days [176,230]. Short-term retinoic acid treatment in this stage enhances the number of iPSC-derived EBs in a dose-dependent manner via induction of both cell proliferation and survival [231]. Differentiated EBs express all genes associated with each of the three germ layers, such as *AFP* or *FOX2* (for endoderm), *SOX1* or β3-tubulin (for ectoderm), and Brachyury or α-smooth muscle actin *(AKTA2)* (for mesoderm) [232,233].

##### Adipogenic Differentiation

The second step of differentiation, consists of culturing iPSC-derived EBs under adherent conditions with an adipogenic cocktail, including FBS, dexamethasone, isobutylmethylxanthin (IBMX), pioglitazone, and insulin [176,186,230,231,234]. In most studies, differentiated EBs are transfected with vectors containing brown or white adipocyte-specific transcription factors and then treated with an adipogenic cocktail. Insulin is a potent adipogenic hormone that stimulates transcription factors responsible for differentiation of pre-adipocytes into mature adipocytes [164]. Moreover, insulin stimulates cells to take up glucose [224]. Insulin promotes not only expression of adipogenic genes (*PPARG* and *FABP4*), but also expression of brown fat genes (uncoupling protein 1 (*UCP1*) and *PGC1A*) [164]. IBMX also increases the expression of thermogenic and adipogenic genes (*PPARG, FABP4, UCP1*, and *PGC1A*) [164,235]. Dexamethasone directly increases the levels of intracellular cyclic adenosine monophosphate (cAMP) and lipolysis in adipocytes [236] and upregulates the expression of *PGC1A, PPARG, FABP4*, and *UCP1* [164]. Using pioglitazone, especially in combination with insulin, results in >95% cell differentiation into lipid-accumulating adipocytes in comparison with 60–80% cell differentiation by treatment with either agent alone [175]. 

##### Differentiation of White Adipocytes from iPSCs

Human iPSC-derived white adipocyte-like cells exhibit lipid accumulation and transcription of adipogenesis-related genes, such as *CEBPA*, *PPARG2*, *LEP*, *FABP4*, and fatty acid-binding protein 4 (*aP2)* [230,231,233,234]. These iPSC-derived adipocytes preserve adipocyte characteristics for 4 weeks after transplantation into mice [234]. Overexpression of *PPARG2*, in combination with adipogenic differentiation medium, is used to enhance differentiation efficiency [176]. Transfected iPSCs show the morphologic appearance of mature white adipocytes even after turning off *PPARG2* transgene expression, and exhibit insulin-stimulated glucose uptake and inducible lipolysis [176]. Without overexpressing adipocyte-specific transcription factors, a method based on retinoic acid treatment for generation of iPSC-derived adipocyte was recently developed [231]; although the functionality of these iPSC-derived adipocyte was not evaluated, the differentiated adipocytes demonstrate higher expression of transcription factors, including FABP4, BODIPY and adiponectin [231]. Patient-specific iPSC-derived adipocytes serve as a platform to investigate diseases related to adipose tissue, e.g., congenital generalized lipodystrophy (CGL), which is characterized by near absence of adipose tissue, severe insulin resistance, hypertriglyceridemia, hepatic steatosis and early-onset diabetes [237]. Lipid-containing cells are barely detectable in iPSC-derived adipocytes from patients with CGL and these cells have lower expression of PPARγ2 [233].

##### Differentiation of Brown Adipocytes from iPSCs

Overexpression of *PPARG2*, *CEBPB*, and *PRDM16* in human [176,238] or mouse [238] iPSCs induces brown adipocytes with the ability of expressing brown adipose tissue markers and releasing glycerol in response to forskolin and β-adrenergic agonists. Differentiated brown adipocytes have multilocular lipid droplets and abundant mitochondria [176,238], high oxygen consumption rate [176,238] and extracellular acidification rate [176], with elevated ATP turnover and proton leak [176], important functional characteristics of brown adipose tissue. Furthermore, transplantation of programmed white and brown adipocytes to immune-compromised mice show a morphology characteristic of primary adipocytes [176]. Transplantation of iPSC-derived brown adipocytes generated from normal or diabetic KK-Ay mice into high fat diet-fed mice results in a lower body weight gain and reduced serum glucose elevation, lower urine glucose, serum levels of total cholesterol, LDL, triglycerides, phospholipid, and non-esterified fatty acids [238]. These results demonstrate that iPSC-derived brown adipocytes are metabolically active after transplantation and suppress diet-induced obesity, dyslipidemia, and T2D.

##### Differentiation of Beige Adipocytes from iPSCs

Adding the Wnt agonist (CHIR99021), recombinant BMP4, VEGFA proteins, and TGF-β inhibitor (SB431542) with an adipogenic cocktail efficiently differentiates FOXF1^+^ splanchnic mesoderm toward beige adipocyte-like cells [239]. Differentiated cells accumulate lipids and express beige adipocyte-enriched genes [239]. Cold exposure leads to secretion of IL-4 from type 2 innate lymphoid cells, which directly stimulates PDGFRα^+^ adipogenic precursors within subcutaneous adipose tissue and increases their differentiation into beige adipocytes [240]. Pretreatment of iPSC-derived MSCs with IL-4 and TGF-β inhibitor leads to a synergistic increase in *UCP1* transcription, as well as beige adipogenic precursor markers (*PDGFRA* and *EBF2*) [239]. Treatment of human iPSCs with two mesodermal inducers (BMP4 and activin A), following an adipogenic cocktail, induces differentiation of beige adipocytes, which store triglycerides and express *UCP1*, *PGC1A*, and *PRDM16*; this characterizes beige and brown adipocytes but not progenitor or mature brown adipocyte markers (*MYF5* or *ZIC1*), suggesting that iPSC-derived cells are not of the brown adipocyte lineage [186]. Activation of a thermogenic program under long-term β-adrenergic stimulation is a key functional feature of beige adipocytes [241]. Accordingly, treatment of iPSC-derived beige adipocytes with a cAMP analog upregulates thermogenic genes, increases intracellular mitochondrial content, and induces shrinking of lipid droplets [186]. Moreover, iPSC-derived beige adipocytes demonstrate a 2.5-fold increase in OCR in response to acute cAMP stimulation [186]. Treating subcutaneous primary adipocytes with iPSC-beige adipocytes derived from T2D patients increases phosphorylation of AKT and glucose uptake in the primary adipocytes upon insulin challenge [239].

**Table 5 cells-09-02465-t005:** Overview of protocols used for differentiation of adipocytes from various sources of iPSCs.

iPSC Source	Protocol	Adipocyte	In Vivo/Vitro Efficacy	Stage/Day	Ref.
Human iPSCs	Days 0–2: Y-27632 and 15%FBS. Days 2–4: 10 µM RA. Days 4–6: 0.1 µM RA. Days 6–7: RA-. Days 7–12: EB plating onto gelatin/matrigel-coated plates. Days 12–20: bFGF. Days 20–30/34: knockout DMEM-F12 containing 10% KSR, Glutamax, IBMX, Dex, Insulin, Indomethacin, and Pioglitazone	White	NR	3/30–34	[231]
Human iPSCs	Days 0–2: Y-27632 and 15% FBS. Days 2–4: 10 µM RA. Days 4–6: 0.1 µM RA. Days 6–7: RA-. Days 7–12: EB plating onto gelatin/matrigel-coated plates. Days 12–20: bFGF. Days 20–30/34: MEM-α supplemented with 10% FBS, IBMX, Dex, Insulin, Indomethacin, and Roziglitazone	White	NR	3/30–34	[231]
Human iPSCs	Days 0–2: 20% KSR. Day 2–5: RA. Day 6–12: 20% KSR. Day 12–22: 10% KSR, IBMX, Dex, Insulin, Indomethacin, Pioglitazone on Poly-L-ornithine and fibronectin plate	White	NR	2/22	[230]
Human and mouse iPSCs	Days 0–3: 10% FBS. Days 3–5: 10% FBS, RA (25 µM). Days 5–7: 10% FBS, RA (50 µM) (floating condition). Days 7–17: 10% FBS, RA (50 µM) (adherent conditions) and overexpression of PRDM16. Days 17–19: 10% FBS, IBMX, Dex, Indomethacin, Insulin, T3, Rosiglitazone. Days 19–27: 10% FBS, Insulin, T3, Rosiglitazone	Brown	↓ Body weight, serum glucose, LDL, total cholesterol, and triglycerides, and urine glucose in high fat diet-fed mice	2/27	[238]
Human iPSCs	Days 0–7: 15% FBS. Days 7–12: 10% FBS, 1% GlutaMAX. Days 12–33: KSR, Dex, hPlasmanate, Insulin, Rosiglitazone	White	Insulin-induced phosphorylation of AKT and glucose uptake. Glycerol release in response to forskolin	3/33	[176]
Human iPSCs	Days 0–2: bFGF. Days 2–5: RA. Days 5–11: Primate ES cell medium. Days 11–14: IBMX, Dex, Insulin, Pioglitazone	White	NR	2/14	[233]
Human iPSCs	Days 0–7: 15% FBS. Days 7–12: 10% FBS, 1% GlutaMAX. Days 12–28: overexpression of PPARγ2, and adding Dox, KSR, Dex, Insulin, hPlasmanate, Rosiglitazone. Days 28–33: KSR, Dex, hPlasmanate, Insulin, Rosiglitazone	White	Insulin-induced phosphorylation of AKT and glucose uptake. Glycerol release in response to forskolin	3/33	[176]
Human iPSCs	Days 0–7: 15% FBS. Days 7–12: 10% FBS, 1% GlutaMAX; Days 12–26: overexpression of PPARγ2, C/EBPα, PRDM16, and adding Dox, KSR, Dex, Insulin, hPlasmanate, Rosiglitazone. Days 26–33: KSR, Dex, hPlasmanate, Insulin, Rosiglitazone	Brown	Glycerol release in response to forskolin and isoproterenol. ↑Oxygen consumption and extracellular acidification rate	3/33	[176]
Human iPSCs	Days 0–2: 20% KSR. Days 2–5: RA. Days 6–8: 20% KSR; Days 8–11: Insulin, Pioglitazone. Days 11–14/16: IBMX, Dex, Insulin, Pioglitazone	White	Insulin-induced phosphorylation of AKT. Glycerol release in response to forskolin	2/14–16	[234]
Human iPSCs	Days 0–4: GlutaMAX, VitC, BMP4, activin A. Days 4–10: 10% FCS, Insulin, IBMX, Dex, Indomethacin. Days 10–20: 10% FCS, Insulin	Beige	Insulin-induced phosphorylation of AKT	3/20	[186]
Human iPSCs	Days −2–0: Serum-free MSC medium, TGF-β inh, IL-4. Days 0–3: Insulin, T3, IBMX, Dex, Indomethacin, TGF-β inh, Rosiglitazone. Days 3–12: Insulin, T3, TGF-β inh, Rosiglitazone;	Beige	Insulin-induced phosphorylation of AKT and glucose uptake	2/12	[239]
Human iPSCs	Days 0–3: 20% KSR. Days 3–5: 20% KSR, RA; Days 6–20: 20% KSR and overexpression of C/EBPβ. Days 20–30: 10% KSR, Insulin, IBMX, Dex, Rosiglitazone	Brown and white	NR	2/30	[242]

KSR, knockout serum replacement; IBMX, 3-isobutyl-1-methylxanthine; Dex, Dexamethasone; RA, Retinoic acid T3, Triiodothyronine; FGF, Fibroblast growth factor; Dox, Doxycycline; FCS, fetal calf serum; NR, Not reported; ↓ decreased.

## 5. Future Prospects and Challenges

Moving the iPSC-derived insulin/glucose-responsive cells from bench to bedside requires overcoming several obstacles. These range from iPSC reprogramming to transplantation of differentiated cells, as well as maintenance of phenotypic stability, and avoiding tumor formation.

### 5.1. Reprogramming of iPSCs

Most iPSCs are established utilizing retroviral vectors. There are numerous transgene integrations in the genomes of iPSCs developed by retroviral vectors, which may cause leaky expression that can interrupt the function of endogenous transcription factors or introduce mutations that increase tumorigenic risk after transplantation. Non-viral reprogramming methods reduce toxicity or other adverse effects of the vectors and their delivery systems, but their efficiency of iPSC induction is typically lower than that of retroviral vectors, possibly due to low transduction efficiency and unstable expression. Furthermore, all reprogramming strategies influence and impede cell cycle control genes, such as p53.

Variations in the iPSC lines strongly influence iPSC reprogramming and differentiation. Embryonic and tail-tip fibroblasts in mouse and skin fibroblast and peripheral blood cells in human are widely used for reprogramming, mainly due to their availability and relative ease of culture. Mouse tail-tip fibroblast iPSCs demonstrate high tumorigenic propensity, while that of gastric epithelial- and hepatocyte-derived iPSCs is lower [243]. This indicates that the origin of the iPSCs influences the risk of tumor formation. A comprehensive study, using 28 iPSC lines derived from various somatic cells, demonstrated that the origin of cells, but not the derivation method, is the major determinant of variation in differentiation [244]. The iPSC clones derived from peripheral blood cells exhibit a high differentiation efficiency, while iPSC clones from skin fibroblasts demonstrate poor differentiation [244]. When comparing iPSCs from peripheral blood and skin fibroblasts from the same individuals, variations in cell differentiation are mainly due to donor differences, rather than to the original cell types [244]. The molecular mechanisms underlying this phenomenon are not yet fully understood, but it has been reported that in iPSCs derived from fibroblasts, mitochondrial numbers return to their pre-reprogrammed state, or even to lower levels, after redifferentiation [245]. In addition, epigenetic imprints of the somatic cell of origin are maintained in the iPSCs, affecting directed differentiation [246].

### 5.2. Immaturity of the iPSC-Derived Insulin/Glucose-Responsive Cells

Despite an increasing number of differentiation protocols, production of mature and functional iPSC-derived insulin/glucose-responsive cells is yet to be convincingly achieved. The final percentage of differentiated β-cells obtained in most studies without purification is at best 30–60%. Most remaining cells are undifferentiated/unwanted cells. iPSC-derived β-like cells demonstrate increased basal insulin levels and insufficient glucose responsiveness. Furthermore, in vitro differentiation results in generation of heterogeneous cell populations, composed of bihormonal (insulin^+^/glucagon^+^) cells alongside diverse categories of progenitor cells [53]. Similarly, most iPSC-derived hepatocytes/myotubes/adipocytes resemble more fetal and neonatal cells. These pieces of evidence indicate lack of additional in vivo factors in in vitro differentiation schemes. For example, after birth, β-cell function changes from amino acid- to glucose-dependent insulin secretion, which reflects the change in the nutritional environment. This adaptation is mediated by a shift in the sensitivity of mammalian target of rapamycin complex 1 (mTORC1) pathway to nutrients [247], which maintains the immature phenotype of the β-cells [248]. Incubation of human fetal β-cells in a mature-like environment (low amino acid concentrations) for four days, leads to a shift into glucose-responsive insulin-secreting cells; incubation of these cells in an immature-like environment (high amino acid concentrations) elicits insulin secretion in response to amino acids but not to glucose [247]. Therefore, growth factors and signaling molecules involved in the development of the pancreas, liver, skeletal muscle, and adipose tissue need to be better characterized to determine their potential to drive PSCs into differentiation of mature insulin/glucose-responsive cells in vitro.

Several modifications in recent differentiation protocols have been made to produce larger populations of mature insulin/glucose-responsive cells. Inhibition of the endogenous Wnt signaling in late stages potentially promotes the differentiation of iPSC-derived β-like cells toward a more mature phenotype [53]. Although Wnt signaling elicits different effects throughout the development of the pancreas [9], this signaling pathway seems to be more important during the initial stage than at the later stages of differentiation. It is required for promoting FGF expression, expansion of the pancreatic epithelium prior to differentiation, and also for terminal differentiation of acinar cells. 

The circadian clock is yet another factor that should be considered when differentiating glucose/insulin-responsive cells in vitro. Entrainment to circadian feeding/fasting rhythms triggers maturation and function of PSC-derived β-cells, evidenced by rhythmic insulin responses and a rise in the glucose threshold for insulin secretion [93]. Circadian rhythms promote expression of enzymes involved in energy metabolism, including glucose transport and metabolism, TCA enzymes, as well as electron transport chain and ATP synthase components [93]. Circadian entrainment also induces maturity-linked factors along with the machinery involved in insulin secretion (secretogranins, synaptotagmins, and syntaxins) [93]. Rhythmic expression of these effectors is associated with greater/pulsatile GSIS responses [93].

Terminal differentiation seems to be feasible by combining factors promoting maturation and 3D differentiation systems. Cell-to-cell surface contacts and simultaneous paracrine signaling regulate cell differentiation. With culture of iPSCs in the presence of MSCs or human umbilical vein endothelial cells (HUVECs), paracrine soluble factors secreted by MSCs or HUVECs, even without cell-cell surface contact, are sufficient to induce hepatic differentiation [140]. However both cell-cell surface contact and paracrine soluble factors must co-exist to allow for the organization into a 3D liver organoid [140]. When both cell-cell surface contact and paracrine signals are in operation, the liver organoids demonstrate a gene expression profile similar to that of primary hepatocytes [140]. In co-culture combinations, mRNA levels of some hepatic markers are, however, lower than in primary hepatocytes, but albumin secretion is ∼2.5-fold higher than hepatocyte-like cells induced from iPSCs alone [140].

Similarly, using a 3D culture system and a step-wise differentiation protocol, pancreatic clusters have been produced, containing α- and β-like cells, and enterochromaffin-like cells, as well as non-endocrine cell type [102]. Using enzymatic dissociation followed by re-aggregation after generation of endocrine progenitors to deplete non-endocrine cells, PSC-derived clusters, which contain up to 80% β-like cells were generated [102]. These highly purified PSC-derived clusters are glucose-responsive, and show elevated stimulation indices compared to unsorted, re-aggregated PSC-derived clusters in both static and dynamic GSIS—but still exhibit lower magnitude of secretion compared to cadaveric islets [102]. However, it must be noted that fractionating the differentiated pancreatic organoids into individual cells to obtain only β-cells would disrupt the architecture of the in vitro generated islets that is sought to recapitulate in vivo development, as cell-cell and cell-matrix contacts are crucial for β-cells to conferring its functional properties.

Standardization of iPSCs meets more difficulty when using patient-specific iPSCs. Insulin mutations impair differentiation of β-cells in a patient-derived iPSC model of neonatal diabetes and triggers ER-stress concomitantly with insulin expression [133]. Increased ER-stress causes reduced mTORC1 signaling and perturbs mitochondria, which are critical for β-cell proliferation and function [133]. Similarly, the differentiation of T1D iPSCs into insulin-producing β-cells exhibits lower efficiency compared to that of non-diabetic iPSCs, and differentiated cells express PDX1 poorly [91]. Effective differentiation of T1D iPSCs requires precise temporal modulation of demethylation; 5-Aza-2′-deoxycytidine (5-Aza-DC), a potent demethylating agent, which inhibits DNA methyltransferase, allows for the binding of the transcriptional machinery and promotes gene expression [91]. In addition, the use of a demethylation agent induces definitive endoderm cells expressing pancreatic β-cell-specific markers, and possessing insulin granules at similar levels to cadaveric β-cells; these cells are functional and glucose-responsive [91]. Furthermore, depending on the cell origin or derivation method, some iPSC lines show resistance to differentiation, variations in differentiation efficiency, or tumorigenicity. These variations seem to be due to epigenetic differences, variations in techniques used to generate or maintain iPSCs, or abnormalities in karyotype. Thus, choosing iPSC lines with minimal batch-to-batch variation in differentiation efficiency is required to overcome this differentiation challenge prior to use in specific cell therapies.

### 5.3. Low Efficiency of Engraftment and Safety in Clinical Therapy

Successful survival post-transplantation is another challenge, which can be attributed to the lack of suitable transplantation approaches and immune rejection in the host. To date, iPSC-derived insulin/glucose-responsive cells in vitro lack many properties of mature cells. Further maturation occurs in vivo after transplantation to immunodeficient mice. Therefore, more work is needed to understand what factors in the in vivo milieu are crucial for functional maturation. Factors may be provided by the in vivo niche at the transplantation site. Maturation develops more efficiently in female mice, indicating the potential role of estrogen in maturation [105]. While human iPSCs can be prepared from the patients themselves, it is still important to develop criteria for cells that can be used clinically. For example, the use of genetic material for reprogramming is still a concern. A recent approach to address this issue is the use of small and large molecules to generate protein- and chemically induced iPSCs, allowing the generation of “nonintegrated” iPSCs (reviewed in [249]). Immunogenicity can be overcome by using feeder-free culture systems, encapsulation of iPSCs, and also by use of xeno-free components in the media [250]. Encapsulation of iPSC-derived cells by semipermeable membrane devices further differentiates cells into mature cells, enhances survival from host immune responses, and reduces or eliminates the need for immunosuppressive agents [14,251,252]. In addition, the CRISPR-Cas9 system for gene editing enables the ablation of the highly polymorphic HLA class I and class II molecules, as well as expressing immunomodulatory factors, such as PD-L1, HLA-G, and CD47 [253,254]. The generation of these hypoimmunogenic human PSCs create “off-the-shelf” products for more wide-spread use.

After satisfying all these requirements, PSC-based therapies may show superiority to any other antidiabetic treatment strategies. Increasing medical expenditures, as a result of the rising prevalence of diabetes, will become an economic concern in the near future. Thus, prevention and treatment of diabetes are very important from a societal perspective. Indeed, restoring or maintaining the function of residual insulin/glucose-responsive cells prevents the progression of diabetes and reduces medical costs.

## 6. Conclusions

Generation of iPSC-derived insulin/glucose-responsive cells in vitro offers great opportunities for both in vitro disease modeling and treatment of diabetes in humans. There is continuous improvement of the differentiation protocols for ESCs and iPSCs, allowing for use in scalable in vitro generation of insulin/glucose-responsive cells, as well as in approaches for transplantation. Despite these advances, a more complete understanding of in vitro differentiation processes and maturation in vivo is required to generate properly functional and safe cells for replacement therapy. This is a pre-requisite for generation of large and sufficient quantities of insulin/glucose-responsive cells, and will also refine and improve in vitro disease models derived from iPSCs.

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
