# Peer review of "Insulin/Glucose-Responsive Cells Derived from Induced Pluripotent Stem Cells: Disease Modeling and Treatment of Diabetes"

_cells, 2020, doi:10.3390/cells9112465_

Round 1

Reviewer 1 Report

The manuscript no. cells-953448 is a very comprehensive, ordered, well-structured and well-documented review paper on insulin/glucose-responsive cells derived from induced pluripotent stem cells. Collecting all the information presented in the manuscript required from the Authors a really tedious work that deserves recognition.

Below some minor remarks have been listed concerning the contents of the manuscript:

In the title of the manuscript two words (i.e. “cells” and “disease”) are mistakenly connected. The part of the title ending with “stem cells” should be separated from the part starting with “disease”.

In Abstract (page 1, lines 10-11) the meaning of the first sentence is unclear: “Type 2 diabetes, (…), accounts for more than 90% of all diabetic patients.” Type 2 diabetes may account for more than 90% of all cases of diabetes, but it cannot not account “for more than 90% of all diabetic patients”.

On page 1 (lines 26-31) the data presented in the first paragraph should be replaced with the most actual data presented in the Eight Edition of the Diabetes Atlas. Consequently, the reference [1] should be updated.

On page 4 (line 185) “for” is missing between “responsible” and “increased”.

An addition of a reference or references at the end of the following statements is suggested: page 6 (lines 240-241), page 7 (lines 288-290.

On page 6 (lines 269-269) the meaning of “pancreatic endocrine fate” is not quite clear.

In subsection 5.2 some information regarding how to obtain functional pancreatic islands with iPSCs-derived insulin-producing cells would be desirable (similar to information provided in lines 710-720

Throughout the manuscript in some symbols / abbreviations normal letters are used while in others - italics. It should be unified.

Author Response

The manuscript no. cells-953448 is a very comprehensive, ordered, well-structured and well-documented review paper on insulin/glucose-responsive cells derived from induced pluripotent stem cells. Collecting all the information presented in the manuscript required from the Authors a really tedious work that deserves recognition.

Response: We thank the reviewer for the encouraging and positive remarks. 

Below some minor remarks have been listed concerning the contents of the manuscript:

Comment: In the title of the manuscript two words (i.e. “cells” and “disease”) are mistakenly connected. The part of the title ending with “stem cells” should be separated from the part starting with “disease”.

Response: This error was created by the on-line system when generating a PDF for the reviewers. This is how the correct title should appear: “Insulin/glucose-responsive cells derived from induced pluripotent stem cells; disease modeling and treatment of diabetes”.

Comment: In Abstract (page 1, lines 10-11) the meaning of the first sentence is unclear: “Type 2 diabetes, (…), accounts for more than 90% of all diabetic patients.” Type 2 diabetes may account for more than 90% of all cases of diabetes, but it cannot not account “for more than 90% of all diabetic patients”.

Response: This has been corrected (Page 1 lines 10-11) as follows: “Type 2 diabetes, characterized by dysfunction of pancreatic β-cells and insulin resistance in peripheral organs, accounts for more than 90% of all diabetes.”

Comment: On page 1 (lines 26-31) the data presented in the first paragraph should be replaced with the most actual data presented in the Eight Edition of the Diabetes Atlas. Consequently, the reference [1] should be updated.

Response: The data have been updated from the 9th edition of Diabetes Atlas: 463 million people have diabetes and 374 million exhibit impaired glucose tolerance; it has been estimated that ~ 578 million people by the year 2030, and 700 million by the year 2045 will suffer from diabetes (Page 1 lines 25-27).

Comment: On page 4 (line 185) “for” is missing between “responsible” and “increased”.

Response: For has been inserted (Page 5 line 212).

Comment: An addition of a reference or references at the end of the following statements is suggested: page 6 (lines 240-241), page 7 (lines 288-290.

Response: References 84-87 (Page 8 lines 301-302) and 114 (Page 10 line 376) have been added.

Comment: On page 6 (lines 269-269) the meaning of “pancreatic endocrine fate” is not quite clear.

Response: According to the comment of reviewer #2, the section “Impact of cellular microenvironment in differentiation of iPSC-derived β-cells” has been entirely edited and this unclear sentence was removed. (Pages 8-9 lines 311-362).

Comment: In subsection 5.2 some information regarding how to obtain functional pancreatic islands with iPSCs-derived insulin-producing cells would be desirable (similar to information provided in lines 710-720

Response: New information regarding differentiation of iPSC-derived pancreatic clusters have been added to the section 5.2 (Page 24-25, lines 829-839; Page 24, lines 791-798; Page 24, lines 809-817), as well as to section 3.2.4 (Pages 6 and 7).

Comment: Throughout the manuscript in some symbols / abbreviations normal letters are used while in others - italics. It should be unified.

Response: We have followed the general guideline for formatting gene and protein names in humans and rodents. In the human, symbols for genes are in upper-case and italicized; protein symbols are identical to their corresponding gene symbols except that they are not italicized. In rodent, gene symbols are italicized, with only the first letter in upper-case. Protein symbols are not italicized, and all letters are in upper-case.

Reviewer 2 Report

In this review article, Gheibi et al. attempted to review the use of iPSCs to model and treat diabetes by generating pancreatic beta cells and insulin target cells (skeletal muscle, adipocytes and hepatocytes). The review tried to summarize the work that has been done on iPSCs only. Unfortunately, in many sections of the review, the authors used old references and neglected the recent ones. To improve this review, there are some major points that should be addressed:

  1. Overall, this review requires extensive revision. It lacks the discussion throughout the manuscript. Also, a clear message showing the recent progress should be added at each section. 
  2. The title should be modified, because it is grammatically not correct.
  3. I believe that “induced pluripotent stem cells” should be replaced by “pluripotent stem cells” to include human embryonic stem cells (hESCs) too, because hESCs have been used as the standard pluripotent stem cells to study pancreatic beta cells and insulin-target cells (hepatocytes, skeletal muscles, and adipocytes). It is known that both types of hPSCs are similar; therefore, most of the protocols and mechanistic studies for modeling and treating diabetes have been done using hESCs.
  4. There The section “2. Reprogramming of somatic cells into iPSCs” should be removed, because it contains repeated details, which are not relevant to the title of the review.
  5. The section title “Differentiation of pancreatic β-like cells from iPSC” lacks key recent information and discussion. Also, it is not well-organized. The authors should discuss the progress in generating functional beta cells.
  6. The authors should discuss which pancreatic progenitors are essential to generate beta cells. The importance of generating progenitors co-expressing PDX1 and NKX6.1. these details are well-presented in the recent review published in CELLS (PMID: 31979403) that should be cited.
  7. Figure 1 will not add any benefits to the readers, because it contains so many cytokines from different publications regardless if those publications generated functional or non-functional beta cells. Therefore, it should be replaced by a Figure to show only the cytokines (from recent publications) that can generate functional beta cells.
  8. Page 6, lines 221-224, the authors cited an old reference (Teo et al. 2012; PMID: 22893457) and mentioned that “activation of NKX6.1 can be initiated before or after endocrine commitment without negative effects on the generation of beta-like cells in vitro”. This is not true statement, because it is well-known in the field that the expression of NKX6.1 in PDX1+ cells before the endocrine commitment is required for generating functional beta cells in vitro and in vivo. I do not why the authors neglected so many recent articles focusing on this point and used old references
  9. Again, the section of “Impact of cell’s physical……..” contains shallow information and mainly cited old references. There are several new publications provided novel findings related to this point. The authors should cite and discuss the following papers: PMID:32094658; PMID: 30487608; PMID: 28824164; PMID: 32857429.
  10. In the insulin resistance section, the authors should discuss recent protocols for generating insulin target cells. Also, is it suitable to use protocols that are established based on genetic modifications? And how these modifications may mask the disease associated phenotypes
  11. Differentiation of while Adipocytes from iPSCs: the authors mentioned that “differentiation of iPSCs to white adipocytes has variable but often low differentiation efficiency (3%-10%). These are not accurate percentages, because recent protocols applied on hESCs and hiPSCs showed higher percentages ranged from 40%-77%. The problem is that the authors always cite old references and do not look for new references. For example, they cited (Taura et al. 2009, PMID:19250937) and neglected CELLS article published in 2020 (Karam et al, PMID:32183164).
  12. Future perspectives: Immaturity of the iPSC-derived…….: the authors did not mention the recent discovery by Melton’s group. They discovered CD49A as a surface marker to purify beta cells from the heterogenous population. In addition, more discussions should be added to the functionality part by citing the recent papers.

Reviewer 3 Report

This review article describes insulin/glucose-responsive cells derived from induced pluripotent stem cells disease modeling and treatment of diabetes. When we consider the usage of clinical therapy, we should prepare xeno-free and nonintegrated hiPSCs or hESCs. Furthermore, universal or hypoimmunogenic hPSCs should be prepared as shelf-of drugs. However, this manuscript did not described these issues. The reviewer suggests to include the importance of preparation method of xeno-free and nonintegrated hiPSCs or hESCs. The following articles should be helpful for the readers if the authors include in the manuscript:

  1. Generation of pluripotent stem cells without the use of genetic material, Laboratory Investigation, 95, 26-42 (2015).

For universal or hypoimmunogenic hPSCs,

  1. Targeted Disruption of HLA Genes via CRISPR-Cas9 Generates iPSCs with Enhanced Immune Compatibility. Cell Stem Cell 24, 566-578 e567 (2019).
  2. Generation of hypoimmunogenic human pluripotent stem cells. Proc Natl Acad Sci U S A 116, 10441-10446 (2019).

a) In Line 35, “reviewed in [3-5]).” The reviewer suggests to include the following review article.

1) Polymeric design of cell culture materials that guide the differentiation of human pluripotent stem cells, Polym. Sci., 65 (2017) 83–126

As for xeno-free culture of hPSCs, the reviewer suggests to include the following review article.

2) Design of polymeric materials for culturing human pluripotent stem cells: Progress toward feeder-free and xeno-free culturing, Prog Polym. Sci., 39(7) (2014) 1348-1374.

b) The following article is important and should be included in this manuscript:

Reversal of diabetes with insulin-producing cells derived in vitro from human pluripotent stem cells. Nat Biotechnol 2014;32:1121–33.

c) The title "induced pluripotent stem cellsdisease modeling" should be revised as "induced pluripotent stem cells disease modeling".

Author Response

This review article describes insulin/glucose-responsive cells derived from induced pluripotent stem cells disease modeling and treatment of diabetes.

Comment: When we consider the usage of clinical therapy, we should prepare xeno-free and nonintegrated hiPSCs or hESCs. Furthermore, universal or hypoimmunogenic hPSCs should be prepared as shelf-of drugs. However, this manuscript did not described these issues. The reviewer suggests to include the importance of preparation method of xeno-free and nonintegrated hiPSCs or hESCs. The following articles should be helpful for the readers if the authors include in the manuscript:

Generation of pluripotent stem cells without the use of genetic material, Laboratory Investigation, 95, 26-42 (2015).
For universal or hypoimmunogenic hPSCs,

Targeted Disruption of HLA Genes via CRISPR-Cas9 Generates iPSCs with Enhanced Immune Compatibility. Cell Stem Cell 24, 566-578 e567 (2019).

Generation of hypoimmunogenic human pluripotent stem cells. Proc Natl Acad Sci U S A 116, 10441-10446 (2019).

Response: We thank the reviewer for these excellent suggestions. We have inserted text, addressing these issues, on Page 25, lines 866-876.      

Comment: In Line 35, “reviewed in [3-5]).” The reviewer suggests to include the following review article: Polymeric design of cell culture materials that guide the differentiation of human pluripotent stem cells, Polym. Sci., 65 (2017) 83–126

Response: The new reference can be found on Page 1, lines 36-37.

Comment:  As for xeno-free culture of hPSCs, the reviewer suggests to include the following review article: Design of polymeric materials for culturing human pluripotent stem cells: Progress toward feeder-free and xeno-free culturing, Prog Polym. Sci., 39(7) (2014) 1348-1374.

Response: The new reference can be found on Page 25, lines 866-876.      

Comment:  The following article is important and should be included in this manuscript: Reversal of diabetes with insulin-producing cells derived in vitro from human pluripotent stem cells. Nat Biotechnol 2014;32:1121–33.

Response: We already refereed to this article in table 2 but the new information from this article has now been added into the main text “Transplantation of iPSC-derived pancreatic PDX1+/NKX6.1+ progenitor cells into diabetic mice reverses hyperglycemia” (Page 5 lines 193-194). And “To provide 3D cell-cell interactions suspension / aggregate / spinner flask culture systems have been developed to induce formation of islet-like organoids which further promote formation of glucose responsive β-cells from stem cells” (Page 9 lines 343-345).

Comment: The title "induced pluripotent stem cellsdisease modeling" should be revised as "induced pluripotent stem cells disease modeling".

Response: This error was created by the on-line system when generating a PDF for the reviewers. This is how the correct title should appear: “Insulin/glucose-responsive cells derived from induced pluripotent stem cells; disease modeling and treatment of diabetes” .

Round 2

Reviewer 2 Report

The authors responded to some of my questions; however, there are still some points to be addressed:

1) If the authors want to keep the section “2. Reprogramming of somatic cells into iPSCs”, they should add more updated information and discuss the reprogramming challenges briefly. Integration-free reprogramming methods should be described, such as the use of the Sendai Virus and episomal vectors as well as other methods. Furthermore, some sentences should be added about the memory of the generated iPSCs and whether these cells have any memory or not and why some of them keep some memory from their original somatic cells.

2) The authors mentioned that they cited CELLS’s review (PMID: 31979403) in pages 4-5, which is not true.

3) Figure 1 still needs modifications, because in its current form does not add clear information. It should be presented differently to be easily understood by the readers. Instead of mentioning the names of all reagents used to generate each stage, it is better to mention each combination that has been previously used or just delete all those cytokines and depend on the Table that should be updated. For example, to generate DE, some protocols use Activin A+ CHIR+ROCK Inhibitor for day 1 and then only Activin A for the following two days. The same should be applied for the rest of stages. Also, some protocols generated beta cells using 7 stage-differentiation protocols, while others like Melton’s protocol used 6 stage differentiation protocol.

Author Response

The authors responded to some of my questions; however, there are still some points to be addressed:

Comment: If the authors want to keep the section “2. Reprogramming of somatic cells into iPSCs”, they should add more updated information and discuss the reprogramming challenges briefly. Integration-free reprogramming methods should be described, such as the use of the Sendai Virus and episomal vectors as well as other methods. Furthermore, some sentences should be added about the memory of the generated iPSCs and whether these cells have any memory or not and why some of them keep some memory from their original somatic cells.

Response: This section was entirely rewritten. Information about integrative and non-integrative reprogramming methods as well as epigenetic memory of iPSCs was added (pages 2 and 3, lines 68-131).

Comment: The authors mentioned that they cited CELLS’s review (PMID: 31979403) in pages 4-5, which is not true.

Response: We apologize for this mistake; it was an error in the reference management software and corrected (page 5, lines 215 and 218, reference 70).

Comment: Figure 1 still needs modifications, because in its current form does not add clear information. It should be presented differently to be easily understood by the readers. Instead of mentioning the names of all reagents used to generate each stage, it is better to mention each combination that has been previously used or just delete all those cytokines and depend on the Table that should be updated. For example, to generate DE, some protocols use Activin A+ CHIR+ROCK Inhibitor for day 1 and then only Activin A for the following two days. The same should be applied for the rest of stages. Also, some protocols generated beta cells using 7 stage-differentiation protocols, while others like Melton’s protocol used 6 stage differentiation protocol.

Response: Small and large molecules used for differentiation of β-like cells were removed from Figure 1 (page 7) and some new information was added to Table 1 (page 11).